

# Low-level isoprene observed during summertime at a forested mountaintop site in southern China: implications for strong regional atmospheric oxidative capacity

Daocheng Gong[1,#], Hao Wang[1,2,#], Shenyang Zhang[1], Yu Wang[3], Shaw Liu[1], Hai Guo[3], Min Shao[1], Congrong He[2,4], Duohong Chen[5], Lingyan He[6], Lei Zhou[1], Lidia Morawska[2,4], Yuanhang Zhang[7], Boguang Wang[1,2,*]

[1]Institute for Environmental and Climate Research, Jinan University, Guangzhou 511443, China
[2]JNU−QUT Joint Laboratory for Air Quality Science and Management, Jinan University, Guangzhou 511443, China
[3]Air Quality Studies, Department of Civil and Environmental Engineering, The Hong Kong Polytechnic University, Hung Hom, Hong Kong
[4]International Laboratory for Air Quality and Health, Queensland University of Technology, GPO Box 2434, Brisbane, Queensland 4001, Australia
[5]State Environmental Protection Key Laboratory of Regional Air Quality Monitoring, Guangdong Environmental Monitoring Center, Guangzhou 510308, China
[6]Key Laboratory for Urban Habitat Environmental Science and Technology, School of Environment and Energy, Peking University Shenzhen Graduate School, Shenzhen 518055, China
[7]State Key Joint Laboratory of Environmental Simulation and Pollution Control, College of Environmental Sciences and Engineering, Peking University, Beijing 100871, China

[#] These authors contribute equally to this work

*Correspondence to*: Boguang Wang (tbongue@jnu.edu.cn)

## Abstract

To investigate the atmospheric oxidizing capacity in certain polluted isoprene-rich environments, such as the forests surrounding megacities. Here we present online observations of isoprene and its first-stage oxidation products methyl vinyl ketone (MVK) and methacrolein (MACR) in summer 2016 at a remote, high-altitude mountain forest site (1,690 m a.s.l.) to the north of the air-polluted Pearl River Delta (PRD) region in southern China. The observed isoprene level was found to be significantly lower in comparison with other forest sites either in China or around the world, although the sampling site was surrounded with subtropical evergreen broad-leaved trees which are strong isoprene emitters. Also, high (MVK+MACR)/isoprene ratio was observed. Based on the observations, we hypothesized that the lower isoprene levels in the study forest might be attributable to a strong atmospheric oxidative capacity in relation to the elevated regional complex air pollution. High daytime OH and nighttime $NO_3$ radical concentrations estimated by using a photochemical box model incorporating Master Chemical Mechanism (PBM-MCM), as well as calculated short atmospheric reaction times of isoprene and long photochemical age, indicated that the isoprene was rapidly and fully oxidized at this aged atmospheric environment, which confirmed our hypothesis. The study suggests that the complex air pollution in the PRD region has significantly



elevated the background atmospheric oxidative capacity of the adjacent forests, and most likely does would probably affect
the regional air quality and ecological environment in the long term. The feedback of forest ecosystems to the increasing
atmospheric oxidation capacity warrants further studies.
**Keywords:** biogenic VOCs; isoprene; atmospheric oxidative capacity; Nanling Mountains; Pearl River Delta
**1 Introduction**
Isoprene is the most abundant non-methane volatile organic compound (NMVOC) in the atmosphere (Guenther et al., 2012).
The reactive chemistry of isoprene affects the oxidation capacity of the troposphere and can contribute to the formation of
ozone ($O_3$) and secondary organic aerosol (Claeys et al., 2004;Lelieveld et al., 2008). The biogenic sources from terrestrial
vegetation contribute more than 90% of atmospheric isoprene, with the largest contribution from forests (Guenther et al.,

11 2006).

Isoprene emissions from forests have been extensively studied over the past decades (Guenther et al., 1991). More recent
work have expanded the focus from emissions to impacts on regional forest chemistry (Seco et al., 2011;Taraborrelli et al.,
2012;Fuchs et al., 2013;Xu et al., 2015;Liu et al., 2016;Kleinman et al., 2016;Su et al., 2016;Gu et al., 2017;Schulze et al.,
2017). These studies have greatly improved our understanding on oxidation process of isoprene, revealed current
uncertainties associated with isoprene emission rates and degradation schemes, and highlighted the biogenic−anthropogenic
interactions in certain polluted isoprene-rich environments, such as the forests surrounding megacities (Hofzumahaus et al.,
2009;Rohrer et al., 2014).
After released into the troposphere, isoprene is rapidly removed mainly by oxidation of hydroxyl (OH) radical and nitrate
($NO_3$) radical. Ambient measurement in pristine forests (e.g. the Amazonian rainforest) found that high OH concentrations
often occur under high-isoprene and low-$NO_x$ ($NO_x \equiv NO + NO_2$, < 1 ppbv) conditions (Lelieveld et al., 2008;Rohrer et al.,
2014). In those areas, OH regeneration contributes greatly to the oxidizing capacity of the atmosphere (Lelieveld et al.,
2008;Fuchs et al., 2013). In addition, relatively high OH-recycling efficiency is not unique to pristine forests, it has been
argued that above an isoprene emitting forest, at high concentration of pollutants, there may be important, but different OH-
recycling mechanism (Hofzumahaus et al., 2009). Studies have also shown that small increases of NO concentration above
the background level can lead to a large change in the air quality of the forest (Liu et al., 2016;Su et al., 2016).
Oxidation by OH radicals dominates daytime isoprene removal, and the oxidation is usually initiated by additional reaction
of an OH across the double bond followed by fast reaction with oxygen ($O_2$). A population of hydroxyl-substituted isoprene
peroxyl radicals (ISOPOO) are thereby produced (Orlando and Tyndall, 2012). The subsequent reactions of ISOPOO
radicals proceed along several competing pathways (Jenkin et al., 2015). In polluted forest areas, the NO pathway likely
dominates. The major products are $NO_2$, methyl vinyl ketone (MVK), and methacrolein (MACR).





The dominant night-time oxidant for isoprene is the $NO_3$ radical (Sobanski et al., 2016;Schulze et al., 2017;Edwards et al.,
2017). $NO_3$ formed through the $O_3$ and $NO_2$ reaction can be abundant at night and react rapidly with isoprene (Brown et al.,
2016;Millet et al., 2016). Production of $NO_3$ directly depends on the mixing ratios of both $O_3$ and $NO_2$. Therefore, in
polluted atmospheres with high levels of $O_3$ and $NO_2$, isoprene oxidation with $NO_3$ is especially important at night.
The major intermediates generated from isoprene oxidation with OH and $NO_3$ are MVK and MACR, which account for
about 80% of the carbon in the initial stage of isoprene oxidation in the atmosphere (Karl et al., 2010). Accurate ambient
measurement of MVK and MACR is a first-order requirement for testing concepts of the reaction pathways of isoprene (Liu
et al., 2016). In addition, measurement of isoprene and its oxidation products provides useful information about the
magnitude and location of isoprene sources (Karl et al., 2009;Guo et al., 2012;Su et al., 2016). Furthermore, diurnal cycles
and compound correlations of isoprene and its oxidation products MVK and MACR at a particular site can yield information
about the locally dominating oxidative agents, such as OH, $NO_3$ and $O_3$ (Guo et al., 2012;Millet et al., 2016), or additional
sources, such as vehicular and industrial emissions (Borbon et al., 2001;Chang et al., 2014;Hsieh et al., 2016).
To deepen the scientific understanding of the biogenic−anthropogenic interactions discussed above, the aim of this study was
to investigate the atmospheric oxidizing capacity by characterizing isoprene and its oxidation products (i.e. MVK and
MACR) at a high-altitude mountain forest site that is highly representative of the upper atmospheric boundary layer in
southern China, a region with large isoprene emissions and strong atmospheric oxidative capacity. A state-of-the-art online
monitoring system was used during a field campaign in summer 2016. To our knowledge, this was the first study of isoprene
observation at a remote, subtropical forested and high-altitude mountain location in southern China.
In this paper, firstly the measured concentration levels and diurnal variations of isoprene and its oxidation products are
presented, then the calculated concentrations of OH and $NO_3$, and finally, the assessment of atmospheric reaction time of
isoprene and the photochemical age of the air mass. Low isoprene levels and high (MVK+MACR)/isoprene ratios were
observed and theoretical calculations confirmed that the rapidly and fully isoprene oxidation might be attributable to a strong
atmospheric oxidative capacity in relation to the elevated regional complex air pollution.
**2 Methods**
**2.1 Site description**
The Pearl River Delta (PRD) region has become one of the most air-polluted areas in China, which happened along with
rapid economic growth and urbanization over the past few decades (Chan and Yao, 2008). It was reported that the OH levels
in the PRD were extremely high, with daily peak values of $1.5-2.6 \times 10^7$ molecules $cm^{-3}$ and nocturnal concentrations within
the range of $0.5-3 \times 10^6$ molecules $cm^{-3}$ (Hofzumahaus et al., 2009;Lu et al., 2012;Lu et al., 2014), significantly higher than
the global mean OH level ($\sim 1.1 \times 10^6$ molecules $cm^{-3}$) (Naik et al., 2013;Lelieveld et al., 2016). The high atmospheric
oxidative capacity has the potential to influence the local and regional air chemistry around the PRD.



To the north of the PRD region lies the Nanling Mountains, an important geographic boundary in southern China separating the temperate areas in the north from subtropical regions in the southeast coast (Siu et al., 2005). The mountain range straddles over 1,400 km from west to east across the borders of four provinces (i.e. Guangxi, Hunan, Guangdong and Jiangxi). The area is the key pathway for the long-range transport of air pollutants from the PRD region to middle-eastern China, particularly during summer, when the southwesterly winds prevail. With a forestry area of 53,600 $km^2$, the Nanling mountain range holds the best preserved and the most representative subtropical evergreen broad-leaved forest in the regions of the same latitude in the world. The trees and shrubs in this subtropical forest are mainly composed of subtropical evergreen broad-leaved trees and Moso bamboo (*Phyllostachys edulis*), both of which are well known to be strong isoprene emitters (Bai et al., 2016a;Bai et al., 2017). Therefore, the Nanling Mountains is an ideal location for studying anthropogenic−biogenic interactions for its high natural emissions and its proximity to anthropogenic pollution sources. So far, however, no isoprene measurements have been conducted in this important area.

The sampling site (24° 41′ 56″ N, 112° 53′ 56″ E, 1,690 m a.s.l.) is located at the summit of Mt. Tian Jing in the centre of Guangdong Nanling National Nature Reserve (24° 37′–24° 57′ N, 112° 30′–113° 04′ E, with an area of 58,368 $hm^2$), southern part of the Nanling Mountains (Fig. 1). Mt. Tian Jing is the highest mountain within a radius of 24 km, with no obstacles around. The site is relatively far from urban and industrial areas, and free of any emissions from local anthropogenic activities; thus serving as one of the national air quality background monitoring stations in China. To the south are the city clusters of the PRD region (200 km north of the metropolitan Guangzhou), which is one of the most urbanized areas in China. During the southwest monsoon (June to September), polluted air from the PRD region or even Southeast Asian may reach the sampling site (Siu et al., 2005;Lin et al., 2017). As the Nanling site is a high altitude mountaintop site in a remote region, and highly representative of the upper atmospheric boundary layer in southern China, measurements of surface isoprene and other species can well represent a large-scale situation.

## 2.2 Measurement Techniques

### 2.2.1 Sampling and analysis of VOCs

The continuous sampling and analysis of ambient VOCs at the Nanling site were conducted automatically by a state-of-the-art online cryogen-free GC−MS system in summer 2016 (*i.e.,* July 15–August 17). The time resolution was 1 h. The VOC measurement instruments were placed inside a two-story building. The sampling tube inlet was located 1.5 m above the rooftop of the building. Ambient air samples were drawn through a 5 m perfluoroalkoxy tube (OD 1/4 inch). The system consisted of a cryogen-free trap pre-concentration device (TH-PKU 300B, Wuhan Tianhong Instrument Co. Ltd., China) and an Agilent 7890A GC/ 5977E MS system (Agilent Tec. USA). The details of this system are described elsewhere (Wang et al., 2014;Li et al., 2016). Briefly, the ambient air was sampled and pumped into an electronic refrigeration and pre-concentration system for 5 min every hour. In order to prevent particulate matter from entering into the sampling system, a Teflon filter was placed in front of the sample inlet. $CO_2$ and moisture were removed by a soda asbestos tube and a water-



removal trap, respectively, before VOC analysis. VOCs were separated on a semipolar column (DB-624, 60 m ×0.25 mm ID
× 1.4 μm, J&W Scientific, USA) and then quantified using a quadrupole MS detector with a full-scan mode.
Rigorous QA (quality assurance) and QC (quality control) procedures were performed through the entire measurement
period. To assess the wall loss of VOCs when passing through the sampling tube, canister sampling at the sampling tube
inlet was conducted simultaneously with the online measurements, and samples were analysed using the offline mode of the
instrument at night of the same day. Twenty-four off-line samples were collected by canisters during the campaign. The
slope and correlation coefficient ($R^2$) of a plot between off-line samples and online measurements for isoprene, MVK and
MACR are 0.98−1.01 and >0.99, respectively. Calibration curves were established for each individual species at seven
different concentrations ranging from 10 to 2,000 pptv before sample analysis. The GC−MS system was also calibrated
using four internal standards (Bromochloromethane, 1,4-Difluorobenzene, Chlorobenzene-d5 and 4-Bromofluorobenzene).
A mixture of 55 non-methane hydrocarbons (NMHCs) and a mixture of oxygenated VOCs (OVOCs) (Linde Electronics and
Specialty Gases, USA) were used to make the certificated curves for calibration. $R^2$ values of calibration curves were >0.99
for all species. Daily calibrations were performed with ±10% variations with reference to the calibration curve results. The
method detection limit (MDL) for isoprene, MVK and MACR quantified with this system was 4, 15 and 10 pptv,
respectively.
**2.2.2 Continuous measurements of trace gases and meteorological parameters**
Ozone ($O_3$) was measured using a commercial UV photometric instrument (model 49i, Thermo Scientific, Inc.), which has a
detection limit of 0.5 ppbv. Oxides of nitrogen ($NO$-$NO_2$-$NO_x$) were measured at 1 min resolution using chemiluminescence
analyser (Thermo Scientific 42i-TL), which has a detection limit of 0.05 ppbv. Sulfur dioxide ($SO_2$) was measured by pulsed
UV fluorescence (model 43i-TLE, Thermo Scientific, Inc.) with a detection limit of 0.05 ppbv. Carbon monoxide (CO) was
monitored using a gas filter correlation infrared absorption trace level analyser (model 48i-TLE, Thermo Scientific, Inc.). A
NIST-traceable standard was applied daily to calibrate the analysers by using Thermo 146i multi-gas calibrator. The zero and
span drift calibrations of the analysers were conducted every two days.
In addition to the above chemical measurements, key meteorological parameters were monitored by an integrated sensor
suite (WXT520, Vaisala, Inc., Finland) including temperature, relative humidity, wind speed, wind direction and
precipitation.
**2.3 OH and $NO_3$ concentrations estimated using photochemical box model**
Since the OH and $NO_3$ concentrations were not measured in this campaign, they were estimated by using a photochemical
box model incorporating Master Chemical Mechanism (PBM-MCM). Since MCM (v3.2) adopts a near-explicit mechanism,
involving 5,900 chemical species and around 16,500 reactions, it has a good performance on calculating free radicals and
intermediate products (Jenkin et al., 1997; Jenkin et al., 2003; Saunders et al., 2003). It is noteworthy that the PBM-MCM
model only considers dry deposition, whereas vertical and horizontal transport is not considered in terms of atmospheric





physical processes. In this study, the observed hourly data of air pollutants ($O_3$, NO, $NO_2$, CO, $SO_2$ and VOCs) and
meteorological parameters (temperature and relative humidity) for the sampling period were input into the model for
simulations. The model output included the averaged concentrations of OH and $NO_3$ radicals. More detailed descriptions of
the PBM-MCM are provided in Ling et al. (2014), Guo et al. (2013) and Cheng et al. (2010).
**2.4 Calculation of the atmospheric reaction time of isoprene**
To calculate the atmospheric reaction time of isoprene, a "sequential reaction approach" based on isoprene's oxidation
mechanism and empirical relationship between isoprene and its oxidation products was used in this study (Stroud et al.,
2001;de Gouw, 2005;Roberts et al., 2006). It is noteworthy that this simplified calculation approach assumes that no fresh
emissions of isoprene are introduced and isoprene emissions are constant during the process. The expression is purely
chemical and does not account for the effects of mixing and transport. We also implicitly assume that the processing time of
the air mass was identical for MVK and MACR and there were no additional sources of MACR and MVK apart from the
oxidation of isoprene. More description about the calculation is given in the Text S1.
**2.5 Photochemical age of the air mass**
Measurements of certain anthropogenic VOCs (e.g. aromatic VOCs) provided us a chance to evaluate the aging degree of the
air mass. Photochemical age is usually used to represent the aging degree of the air mass, and it can be calculated by the
ratios of two VOC species that share common emission sources but with large different reactivities with OH (de Gouw,
2005;Shiu et al., 2007;Parrish et al., 2007;Yuan et al., 2012;Yang et al., 2017). Although mixing of fresh emissions with
aged air masses will introduce substantial uncertainties in the determination of photochemical age, it still provides useful
measures of photochemical processing in the atmosphere. In this study, we chose three pairs of aromatic species:
toluene/benzene, ethylbenzene/benzene, and m,p-xylene/benzene. Details about the photochemical age calculation are given
in the Text S2.
In addition, the open source R package "openair" (Carslaw, 2015;Carslaw and Ropkins, 2012) was utilized for data analysis
and graph plotting (see Text S3).
**3 Results and Discussion**
**3.1 Time series of meteorology and trace gases**
The time series of selected meteorological parameters and trace gases are presented in 1 hour averages (Fig. 2).
Discontinuities in the figure indicate that either no data were available due to the calibration and maintenance of the
instruments or the values were below the MDL for those time periods. During the study, the air masses reaching the site
were mainly from the southwest and northeast directions. With the change of meteorological parameters, the mixing ratios of



air pollutants changed correspondingly. In particular, from July 23 to 27, concentrations of anthropogenic pollutants ($SO_2$,
CO and aromatic VOCs) dramatically increased, and were probably affected by regional transport. During July 28−31, due
to the relatively higher temperature and lower surface wind, the emissions of isoprene were enhanced, and the dispersion of
isoprene and its oxidation products was reduced, resulting in elevated levels of these species in the air. In addition, there was
a notable decrease in concentrations of both isoprene and its oxidation products in August 2−3 caused by continuous rain
during the typhoon NIDA.
The average hourly levels of isoprene, MVK, MACR, benzene, toluene, ethylbenzene and m,p-xylene were $287 \pm 32$ pptv
(4−2605 pptv), $293 \pm 22$ pptv (16−1244 pptv), $73 \pm 6$ pptv (10−442 pptv), $51 \pm 8$ pptv (4−992 pptv), $154 \pm 20$ pptv (19−1770
pptv), $47 \pm 6$ pptv (2−499 pptv) and $38 \pm 4$ pptv (7−274 pptv), respectively. The concentrations of $O_3$, $NO_2$, NO, CO and
$SO_2$ ranged from 14.4 to 130.6 ppbv (mean = 53.5), 0.9 to 10.5 ppbv (mean = 2.4), 0.6 to 8.7 ppbv (mean = 0.7), 40.8 to
684.4 ppbv (mean = 260.3) and 0.5 to 3.1 ppbv (mean = 0.9), respectively. The average temperature was $19.2 \pm 0.1$ ℃ and
the relative humidity was $92.1 \pm 0.6$ %.
Isoprene concentrations vary in locations and seasons due to the difference in forest types, ambient oxidation processes and
related meteorological parameters (e.g. temperature and sunlight). Surprisingly, comparison revealed that the isoprene level
in this study was much lower than that observed at other sites of the same type of forest, either in China or around the world
(Table 1), particularly if considering a fact that potentially strong isoprene emitters, like evergreen broad-leaved trees and
shrubs, are widely seen in this low latitude subtropical-forested region (Bai et al., 2016a;Bai et al., 2017). Although the high-
altitude feature (1,690 m a.s.l.) of this mountain site may lower the observed isoprene levels as compared with the forest
canopy underneath the site, it is interesting to find that the daytime isoprene concentration ($377 \pm 46$ pptv) in the hottest
months (July−August) of the year was 0.5−1.0 times lower than the values observed at the same latitude subtropical-forested
sites in Southern China (e.g. yearly value of 760 pptv at DingHu Mountain, and summer average of 554 pptv in HongKong)
(Chen et al., 2010;Wu et al., 2016), and even slightly lower than the autumn values (410 pptv) of a site (3,250 m a.s.l.)
located on the Tibetan Plateau (Bai et al., 2016b). Furthermore, $O_3$ and $NO_x$ levels at this site were generally higher than the
observations available in other forest studies worldwide (Table 1), likely suggesting the relevance of the low observed
isoprene levels with the complex atmospheric pollution in this region.
**3.2 Diurnal variations**
The diurnal behaviours of isoprene, MVK and MACR are influenced by a number of chemical (e.g. oxidants levels) and
meteorological (e.g. temperature) factors. Fig. 3 shows the average diurnal patterns of isoprene, MVK and MACR. The
diurnal variations of (MVK+MACR)/isoprene ratios, temperature, $O_3$, $NO_2$, NO and CO are also shown in the figure. During
the sampling periods (July 15−August 17, 2016), the sunrise and sunset times were 05:49−06:03 and 18:57−19:15 LT,
respectively. The mixing ratio of isoprene started increasing at 7 a.m., peaked at 2 p.m., and then gradually decreased to a



low level at night, and remained at this level until 7 a.m. of the next day. The levels of isoprene, MVK and MACR decreased
substantially at 6 a.m., likely due to the expansion of the atmospheric boundary layer (ABL) and entrainment of oxidants-
rich free tropospheric (FT) air into the ABL (Vilà-Guerau de Arellano et al., 2011). The hourly averaged daytime
(06:00–18:00 LT) levels of isoprene ($377 \pm 46$ pptv, $p < 0.01$) and MVK ($332 \pm 32$ pptv, $p < 0.01$) were both higher than
their average nighttime values ($159 \pm 35$ pptv and $252 \pm 28$ pptv, respectively). However, the daytime level of MACR ($66 \pm$
$7$ pptv) was slightly lower than its average nighttime value ($81 \pm 10$ pptv).
Isoprene mixing ratios were consistently higher during daytime and lower at nighttime, indicating higher net production of
isoprene during the day compared to the night. Daytime MVK and MACR are formed dominantly from the reaction of
isoprene with OH radical (Reissell and Arey, 2001). The rapid decrease in isoprene after sunset was attributed to the reaction
with $NO_3$ radical (Apel, 2002). In this study, the MVK mixing ratios during the day were higher than those during the night,
suggesting that the MVK was mainly formed from the reaction of daytime produced isoprene with OH. In addition, the
remaining isoprene after daytime photochemical loss reacted with $NO_3$ at night, contributing to the nighttime MVK
formation. Although the yield of MACR and MVK from isoprene $NO_3$-oxidation are the same (0.035) and MACR react
faster with $NO_3$ than MVK (Table S1). Surprisingly, the nighttime MACR levels were slightly higher than those during the
day, probably due to the reaction of daytime residual isoprene with high levels of nighttime $O_3$, as the yield of MACR from
isoprene $O_3$-oxidation is nearly 2 times higher than that of MVK (Table S1). In this study, due to the remote and high-
altitude nature of the site, both daytime isoprene photochemistry and nighttime $NO_3$ chemistry played an important role in
the diurnal patterns of isoprene, MVK and MACR. Interestingly, higher MACR levels at night than that during the day may
attributed to the high nighttime $O_3$.
In this remote forest area, isoprene oxidation was the dominating source of MVK and MACR. During the daytime and
nighttime periods, the (MVK+MACR)/isoprene ratio at a particular location is driven in part by the dominant daytime OH
and nighttime $NO_3$ chemistry, which consumes isoprene while producing and destroying MVK and MACR. The ratio is
expected to depend on factors such as the isoprene emission rate, the $NO_x$-dependent radical concentration, the degree of
atmospheric mixing, and distance from isoprene emitters (Montzka et al., 1995;Biesenthal et al., 1998). In this study, it is
somewhat surprising that despite these effects, the calculated ratio is quite high, averaging $4.0 \pm 0.8$, as shown in Fig. 3. The
ratio at nighttime hours ($6.3 \pm 1.4$, $p<0.01$) is much higher than that ($1.9 \pm 0.5$) during daytime hours. The diurnal pattern of
the ratio of this study is consistent with the results by Biesenthal et al. (1998) and Apel (2002), with both studies showing
higher values during nighttime hours. The average ratio of (MVK + MACR)/isoprene is notably higher than that (0.12 and
2.0 for daytime and nighttime hours, respectively) by Apel (2002), in which sampling site was ~12 m above a rural forest
canopy. The high (MVK + MACR)/isoprene ratios in this study are in close agreement with Kuhn et al. (2007), who reported
an increase of the ratios with the height within the ABL. In addition, studies have shown that enhanced levels of the
(MVK+MACR)/isoprene ratio are expected in environments where the air mass has aged under high-$NO_x$ and high-oxidants
conditions (Apel, 2002). In this study, the site was in a relatively high $NO_x$ and oxidants regime and this may have



contributed to the observed high ratios. This remarkably high (MVK+MACR)/isoprene ratio were indicative of a remarkably
high oxidation capacity, likely suggesting that isoprene was fully oxidized at this site in both daytime and nighttime periods.
**3.3 Estimated concentrations of OH and NO$_3$ radical**
It is well known that OH radical is largely responsible for the daytime isoprene removal while NO$_3$ radical become more
important in the oxidation of isoprene, MVK and MACR at night (Starn et al., 1998;Reissell and Arey, 2001). The diurnal
profiles of model-calculated OH and NO$_3$ radical are shown in Fig. 4.
**3.3.1 Daytime OH**
The average hourly daytime OH concentration estimated by PBM-MCM at this remote forest site was $7.3 \pm 0.5 \times 10^6$
molecules cm$^{-3}$ ($0.36 \pm 0.03$ pptv), with a median value of $7.7 \times 10^6$ molecules cm$^{-3}$. Peaks in concentrations ($14.4 \pm 0.8 \times$
$10^6$ molecules cm$^{-3}$) appeared at 12:00 LT when the solar radiation was usually the strongest, and then gradually the
concentrations decreased to the lowest levels before sunset. The calculated average OH level in this study is consistent with
the results in the PRD region ($\sim 8 \times 10^6$ molecules cm$^{-3}$) (Xiao et al., 2009;Yang et al., 2017;Hofzumahaus et al., 2009). And
the range of estimated mixing ratios of daytime OH ($3.6 \times 10^5$ to $1.9 \times 10^7$ molecules cm$^{-3}$) in this work generally agrees
with the daytime levels (hourly value ranged from $3.3 \times 10^6$ to $2.6 \times 10^7$ molecules cm$^{-3}$) observed by Xiao et al. (2009),
Hofzumahaus et al. (2009) and Lu et al. (2012) at rural site in the PRD region. The modelled daytime peak OH value is
much higher than those observed daytime maxima at remote forest areas such as Blodgett forest in California ($4 \times 10^6$
molecules cm$^{-3}$) (Mao et al., 2012), boreal forest in Finland ($3.5 \times 10^6$ molecules cm$^{-3}$) (Hens et al., 2014), pine forest in
Alabama ($1 \times 10^6$ molecules cm$^{-3}$) (Feiner et al., 2016) and Mount Tai in Central China ($5.7 \times 10^6$ molecules cm$^{-3}$) (Kanaya
et al., 2009). Limited studies performed in the PRD region have confirmed the strong atmospheric oxidizing capacity in the
polluted atmosphere of this region (Hofzumahaus et al., 2009;Lu et al., 2012;Xue et al., 2016;Ma et al., 2017;Li et al., 2018).
The high model-derived concentrations of OH in this study indicate that the atmospheric oxidative capacity of this forested
region was strong, which facilitates fast oxidation of daytime isoprene.
**3.3.2 Nocturnal NO$_3$**
The estimated average nighttime hourly NO$_3$ level was $6.0 \pm 0.5 \times 10^8$ molecules cm$^{-3}$ ($29 \pm 3$ pptv). The estimated levels
of nighttime NO$_3$ for this remote mountain site are comparable to the results ($\sim 40$ pptv) obtained at a semi-rural mountain
site (825 m a.s.l.) (Sobanski et al., 2016) and lower than the levels ($\sim 70$ pptv) observed at a high-altitude (2,280 m a.s.l.)
mountain site (Chen et al., 2011). The NO$_3$ levels in this study were higher than that (11 pptv) modelled by Guo et al. (2012)
and close to that ($\sim 31$ pptv) observed by Brown et al. (2016) both at a mountaintop site (640 m a.s.l.) in Hong Kong. The
mixing ratios of NO$_3$ started steady increasing at 7 p.m., peaked at 8 p.m., then rose gradually after midnight, and peaked
again at 2 a.m. of the next day. The nocturnal variation of NO$_3$ is similar to that of O$_3$ (peak at 8 p.m.). At our study site, the
average nighttime mixing ratios of NO$_2$ ($2.5 \pm 0.1$ ppbv) and O$_3$ ($55.5 \pm 2.1$ ppbv) were relatively high when compared with



other remote forest sites ($NO_2 < 1$ ppbv, $O_3 < 30$ ppbv), providing more favourable conditions for the $NO_3$ formation. In
addition, in the surface layer of urban areas, $NO_3$ is generally low due to the existence of continuously anthropogenic NO as
an important $NO_3$ sink; however, in remote or high-altitude mountain regions with cleaner air aloft (e.g. in the upper ABL or
FT), higher $NO_3$ are often observed (Chen et al., 2011;Sobanski et al., 2016;Wang et al., 2017). The vertical profiles of $NO_3$
(Fish et al., 1999;Friedeburg et al., 2002;Stutz et al., 2004) suggest that the $NO_3$ concentration increases with altitude, with a
significant fraction existing in the upper ABL or FT (Allan et al., 2002). This is consistent with our results obtained at this
high-elevation mountain site. Therefore, the relatively high nighttime $NO_3$ concentrations at this high-altitude mountain site
may lead to fast decay of daytime residual isoprene and consequently contribute to MVK and MACR formation.
**3.4 Atmospheric reaction time of isoprene**
As MVK and MACR are dominant first-generation reaction products from isoprene oxidation, a relationship can be expected
between the concentrations of isoprene and these species (Biesenthal et al., 1997). The ratios of MVK/isoprene and
MACR/isoprene provide useful information on the oxidation process of isoprene in an air mass (Stroud et al., 2001;Apel,
2002;Roberts et al., 2006;Kuhn et al., 2007;Xie et al., 2008;Liu et al., 2009;Guo et al., 2012;Wolfe et al., 2016). In this study,
since the OH and $NO_3$ concentrations were not measured and varies as an air mass ages, we prefer to use the term "exposure"
(Jimenez et al., 2009;Wolfe et al., 2016;de Gouw, 2005), defined here as the product of radical concentration and reaction
time for the isoprene in the atmosphere between emission and detection.
Fig. 5 compares the observed relationship of observed MVK/isoprene and MACR/isoprene ratios against theoretical trends
predicted by the sequential reaction model for the daytime and nighttime hours. It can be seen that the observed ratios of
MVK/isoprene versus MACR/isoprene exhibit a tight linear correlation ($R^2$=0.68 and 0.72 for daytime and nighttime periods,
respectively). The measured data fit the predicted line well, although most of the measured data are above the predicted line,
consistent with the observations of several previous studies (Stroud et al., 2001;Apel, 2002;Guo et al., 2012). This might be
caused by a continuous supply of MVK and MACR from surrounding forest trees during the daytime hours and additional
source from oxidation by daytime $NO_3$ and nighttime OH (Brown et al., 2005;Faloona et al., 2001), which were not taken
into account in the sequential reaction modeling. The theoretical slope agrees well with observations, indicating exposures of
$0.1-12 \times 10^6$ OH $cm^{-3}$ h and $4-28 \times 10^8$ $NO_3$ $cm^{-3}$ h for daytime and nighttime periods, respectively. For a typical daytime
average OH concentration of $8 \times 10^6$ molecules $cm^{-3}$ (Xiao et al., 2009;Yang et al., 2017;Hofzumahaus et al., 2009) and
nighttime average $NO_3$ concentration of $5 \times 10^8$ molecules $cm^{-3}$ (Guo et al., 2012;Brown et al., 2016), this corresponds to
daytime and nighttime processing times of 0.01–1.5 h and 0.8–5.6 h, respectively.
Exposures can be calculated from observed daughter/parent ratios. Fig. 6a shows the derived exposures from MVK/isoprene
and MACR/isoprene ratios. Calculated daytime OH exposures and nighttime $NO_3$ exposures range from $1.0 \times 10^5$ to $1.3 \times$
$10^7$ molecules $cm^{-3}$ h and $3.5 \times 10^8$ to $3.2 \times 10^9$ molecules $cm^{-3}$ h, respectively. OH and $NO_3$ exposures derived from two
methods exhibit a good linear correlation ($R^2$=0.63 and 0.70 for OH and $NO_3$, respectively), and results derived from MACR
are 4% and 18% lower than those from MVK on average, respectively, and we use the mean of these two values. The




median and mean OH exposure is 1.9 and $2.5 \times 10^6$ molecules $cm^{-3}$ h, respectively. For $NO_3$ exposure, the median and mean
value is close (15.8 and $16.2 \times 10^8$ molecules $cm^{-3}$ h, respectively). The mean daytime and nighttime isoprene reaction time
are 0.3 and 3.2 hours, respectively, assuming daytime OH = $8.0 \times 10^6$ molecules $cm^{-3}$ and nighttime $NO_3 = 5 \times 10^8$
molecules $cm^{-3}$. The isoprene processing time is mainly relevant to OH and $NO_3$ mixing ratios, which varied spatially and
temporally, and the proximity to isoprene sources.
To obtain the detailed profiles of the isoprene atmospheric reaction time at the site, we calculated them which based on the
modelled OH and $NO_3$ results in this study. Fig. 6b shows the derived reaction times from MVK/isoprene and
MACR/isoprene ratios. Reaction times derived from two methods exhibit a significant linear correlation ($R^2$=0.91 and 0.90
for daytime and nighttime, respectively), and results derived from MACR are 13% lower than those from MVK on average,
and we use the mean of these two values. The calculated isoprene reaction time during the day is between 0.01 and 14.43
hours, with median and mean values of 0.27 and 1.39 hours, respectively. The isoprene reaction time during the night was
calculated to be between 0.30 and 16.44 hours, with median and average values of 4.10 and 4.49 hours. The longer isoprene
reaction time at night than during the day is probably due to the lower reaction rate of isoprene with $NO_3$ than with OH. The
daytime residual MVK and MACR after sunset may also have significant impacts on the calculated nighttime reaction time,
as the life time of MVK and MACR for reaction with $NO_3$ is long (0.5 years and 72 hours at a 12-h nighttime average $NO_3$
of $5.0 \times 10^8$ molecules $cm^{-3}$, respectively). The median daytime reaction time (0.27 hours) of measured isoprene was slightly
lower than the theoretical lifetime of isoprene (0.4 hours at 12-h daytime averaged [OH] = $8.0 \times 10^6$ molecules $cm^{-3}$). In this
study, the average distance between the sampling site and the centre of the emitting trees was about 20 km. The daytime
reaction time of isoprene (16 min) in this study is lower than that (~30 min) of Guo et al. (2012) in which the sampling site
was 5 km away from the centre of the large forests in Hong Kong. And that means the short reaction time of isoprene in this
study was probably attributed to the high oxidants levels.
**3.5 Initial mixing ratios of isoprene**
To check out the magnitude of isoprene oxidation, "initial isoprene", the total isoprene emissions that have been released
into the sample air masses, can be effectively calculated via reverse integration of isoprene's first-order oxidation (Wolfe et
al., 2016):
$[ISOP]_0 = [ISOP] \times e^{(k \times EXPO)},$                         (1)
Where $[ISOP]_0$ is the initial isoprene, representing the amount of isoprene that an air parcel would have to start with to
generate the amount of isoprene, MVK and MACR observed. $[ISOP]$ is the observed isoprene. $k$ is the reaction rate
coefficients for the reactions of isoprene with OH and $NO_3$ radical. $EXPO$ is the calculated OH and $NO_3$ exposures.
Fig. 7a shows plots of the initial isoprene versus the observed isoprene. The daytime initial isoprene mixing ratios (1213 ±
108 pptv) is much higher than the observed values (377 ±46 pptv). It is noteworthy that the nighttime initial isoprene by this
approach may be overestimated due to the daytime residual MVK and MACR into the night. The daytime initial mixing
ratios of isoprene are 1–20 times higher than the observed levels, with median and mean values of 2.1 and 4.3, respectively.





Scatter plots of calculated initial isoprene versus measured MVK+MACR during daytime hours are also given in Fig. 7b,
and a good correlation ($R^2$=0.71) was obtained. Since the slope is related to the yield of (MVK+MACR) from OH-initiated
reaction of isoprene and further oxidation of those two products with OH, data points away from the dashed line are likely
due to chemical loss of MVK and MACR and/or the influence of continuous emissions of isoprene. These results further
confirmed that isoprene was fully oxidized in the air masses.
**3.6 Aging degree of the air mass**
Fig. 8 shows the calculated photochemical age (PA) from daytime toluene/benzene (referred to as T/B),
ethylbenzene/benzene (E/B), and m,p-xylene/benzene (X/B) ratios. PA derived from the three methods exhibit a good linear
correlation ($R^2$=0.82 and 0.79 for $PA_{T/B}$ versus $PA_{X/B}$ and $PA_{E/B}$ versus $PA_{X/B}$, respectively). Results derived from X/B are 37%
and 24% lower than those from T/B and E/B on average, respectively, and we use the mean of these three values in this
study. The higher mean values than median values for all methods indicating certain impacts of outflow from urban areas
(e.g. the PRD region) (Suthawaree et al., 2012), when the polluted air mass arriving from those areas transported to the site
leads to higher photochemical age (1.4−8.2 days). The median and mean PA are 3.8 and 12.4 h, respectively. The average
PA in this study was about twice times of the observations (6−7 hours) in a suburban site in the PRD region (Yang et al.,
2017), indicating a more aged atmospheric environment in this remote site.
**4 Conclusions**
In this study, isoprene and its major intermediate oxidation products MVK and MACR were simultaneously observed in
real-time in 2016 summer season at a high-altitude mountain forest site located at the Nanling Mountains in southern China.
Although the sampling site was surrounded with subtropical evergreen broad-leaved trees which are strong isoprene emitters,
the observed isoprene level (377 $\pm$ 46 pptv) was found to be significantly lower than other remote forest studies, while
(MVK+MACR)/isoprene ratio (4.0 $\pm$ 0.8) was relatively higher. Based on the observations, we hypothesized that the lower
isoprene levels in the study forest might be attributable to a strong atmospheric oxidative capacity in relation to the elevated
regional complex air pollution.
To validate this hypothesis, high daytime OH and nighttime $NO_3$ radical concentrations were estimated by using a PBM-
MCM, with average hourly mixing ratios of 7.3 $\pm$ 0.5 $\times 10^6$ (0.36 $\pm$ 0.03 pptv) and 6.0 $\pm$ 0.5 $\times 10^8$ (29 $\pm$ 3 pptv) molecules
cm$^{-3}$, respectively. The modelled values are comparable to those observations conducted in the adjacent PRD region. The
high model-derived radical levels indicate the strong atmospheric oxidative capacity in this subtropical-forested region,
which facilitates fast isoprene oxidation and subsequently contributes to the MVK and MACR formation.
In addition, the term "exposure" was used to express the isoprene processing, with mean daytime OH and nighttime $NO_3$
exposure of 2.5 $\times 10^6$ and 16.2 $\times 10^8$ molecules cm$^{-3}$ h was obtained, respectively. Short atmospheric reaction times of
isoprene during the day (0.27 h) and night (4.10 h) were subsequently calculated based on the estimated radical



concentrations. Also, the initial isoprene was 4.3 times higher than the observed isoprene, and the photochemical age (12.4 h)
at this site was about twice times of that in the PRD region. These indicate that the isoprene was rapidly and fully oxidized at
this aged atmospheric environment.
To the best of our knowledge, there are no direct measurements of isoprene and its first-stage oxidation products at this
remote, subtropical forested and high-altitude mountain location in southern China; thus, the results presented here constitute
the first measurement-constrained evaluation of the early-stage isoprene oxidation. In this regard, the current work has
highlighted that the air quality and ecological environment of this forest was affected by the highly polluted air in the PRD
region and has led to enhanced oxidation capacity of the forest′s atmosphere. Continued field observations and further
studies are crucial for understanding the relatively high oxidative capacity of this region and for exploring the feedback of
forest ecosystems to the increasing atmospheric oxidizing conditions.
**Acknowledgments**
This work was supported by the National Natural Science Foundation of China (91544215, 41373116). The authors thank Jie
Ou, the chief engineer of Shaoguan Environmental Monitoring Central Station, for the help during the sampling campaign.
We also acknowledge Dr. David Carslaw for the provision of the R package "openair" (http://www.openair-project.org) used
in this publication. We also thank the Team BlackTree for providing an aerial photo of the Nanling site in Fig. 1c.

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





1  **Tables**

2  **Table 1: Comparison of average concentrations (ppbv) of isoprene, $O_3$, NO and $NO_2$ measured at the Nanling site, as well as**
3  **(MVK+MACR)/isoprene ratios (ppbv/ppbv), with other remote forest sites.**

| Forest type and latitude | Isoprene | Ratio | $O_3$ | NO | $NO_2$ | Sampling time | References |
|---|---|---|---|---|---|---|---|
| Subtropical (24.70° N) | 0.377 | 1.9 | 51.9 | 0.803 | 2.386 | Daytime (Jul. – Aug.) | This study |
| | 0.159 | 6.3 | 55.5 | 0.656 | 2.511 | Night (Jul. – Aug.) | |
| Subtropical (23.17° N) | 0.760 | - | - | - | - | Daytime (All year) | (Wu et al., 2016) |
| Subtropical (22.29° N) | 0.554 | - | - | - | - | Daytime (Summer) | (Chen et al., 2010) |
| Tropical (18.40° N) | 0.480 | - | - | - | - | Daytime (Apr.) | (Tang et al., 2007) |
| Deciduous (22.25° N) | 0.370 | - | 30.0 | - | - | Daytime (All year) | (Wang et al., 2005) |
| Temperate (42.40° N) | 1.720 | - | - | - | - | Daytime (all year) | (Wang et al., 2008) |
| Tibet (37.59° N) | 0.410 | - | - | - | - | Daytime (Sep.–Oct.) | (Bai et al., 2016b) |
| Temperate (45.56° N) | 1.360 | 0.1 | - | 0.1 | 1.000 | Daily (Summer) | (Apel, 2002) |
| Tropical (4.98° N) | 1.058 | 0.5 | - | - | - | Daily (Apr.–Jul.) | (Jones et al., 2011) |
| Deciduous (36.21° N) | 0.743 | 0.6 | - | - | - | Daily (Jun.–Jul.) | (Link et al., 2015) |
| Coniferous (38.90° N) | 0.397 | 2.3 | - | - | - | Daily (Jun.–Sep.) | (Dreyfus et al., 2002) |
| Oak (45.20° N) | 1.070 | 0.5 | - | - | - | Daily (Jun.–Jul.) | (Acton et al., 2016) |
| Mediterranean (41.78° N) | 0.430 | 0.7 | 37.5 | 0.8 | 1.000 | Daily (Jul.–Aug.) | (Seco et al., 2011) |
| Tropical (2.59° S) | 1.660 | - | - | - | - | Daytime (wet season ) | (Alves et al., 2016) |
| Tropical (2.59° S) | 3.400 | 0.31 | 15.0 | - | - | Daytime (dry season) | (Kuhn et al., 2007) |



**Figures**

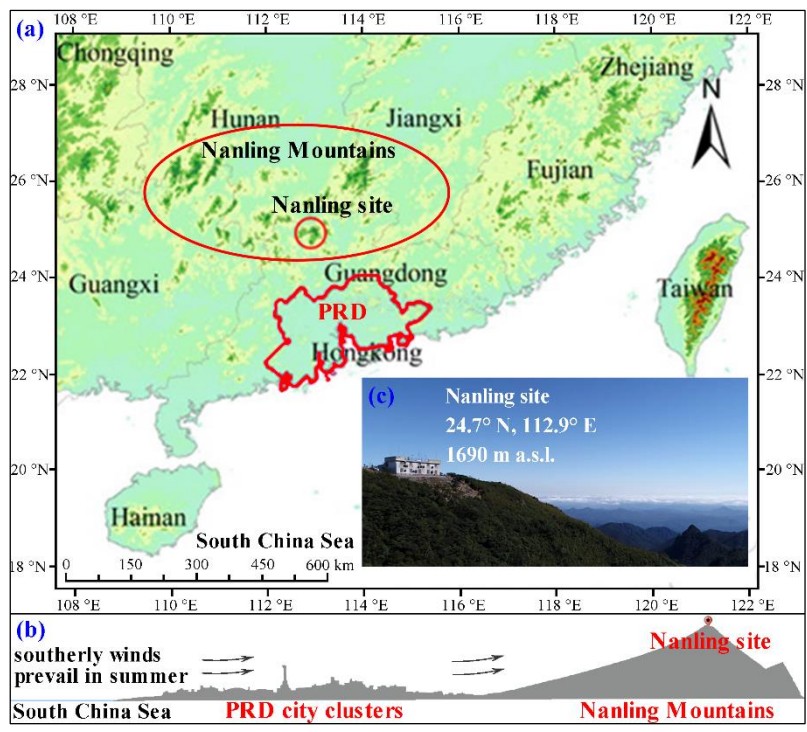

Fig. 1: (a) Map showing the location of the Nanling site at the summit of Mt. Tian Jing in southern Nanling Mountains; (b) Also
shown is the sketch of cross section of the PRD and Nanling Mountains; (c) Aerial photo of the Nanling site. The base map in Fig.
1a and Fig.1b is reproduced from Wu et al. (2016) and Wu et al. (2013), respectively.







Fig. 2: Time series (1 hour data) of trace gases and meteorological parameters during July−August 2016 at the Nanling site. Blue
dashed lines are Grade I of the Ambient Air Quality Standard in China for O₃ (80 ppbv). Temperature, relative humidity, wind
speed and wind direction are referred to as Temp., RH, WS and WD, respectively.



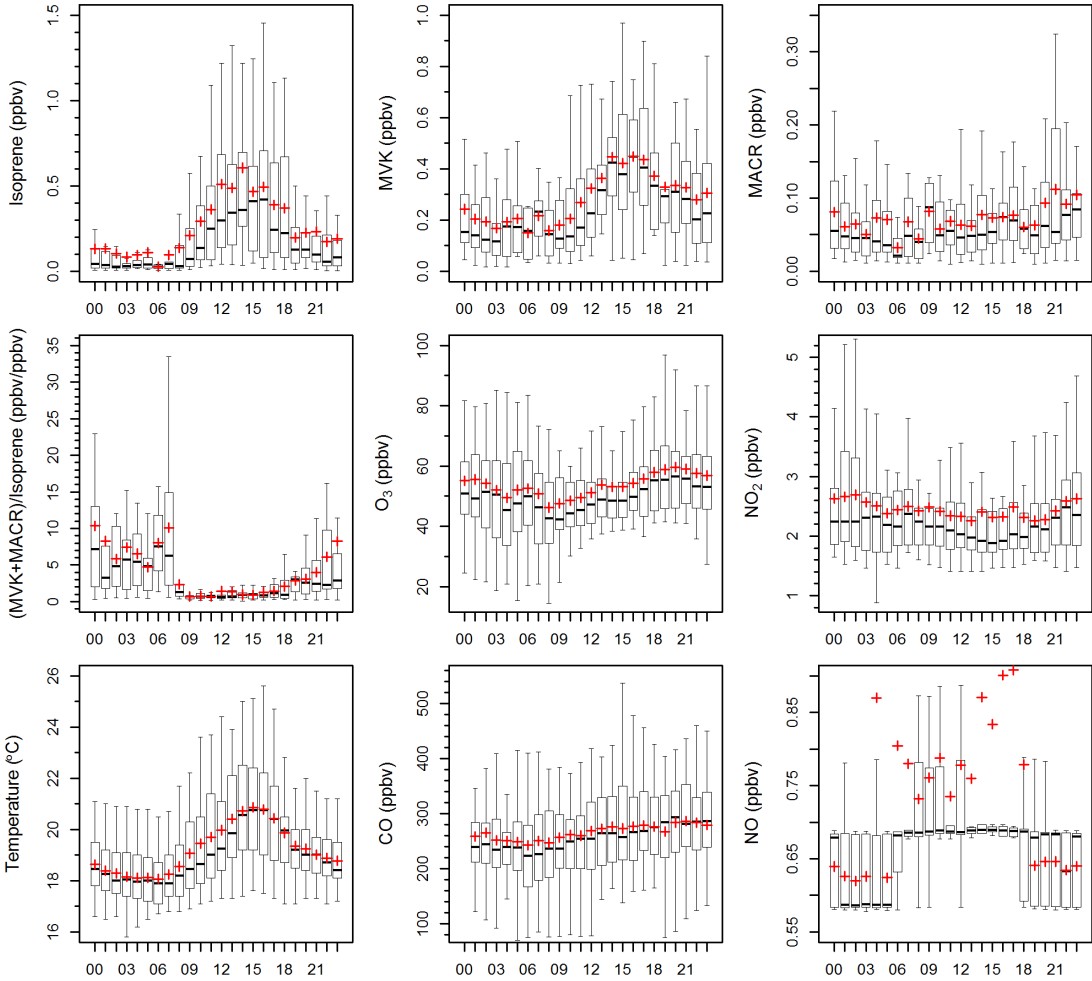

Fig. 3: Box and whisker plots of average diurnal patterns of isoprene, MVK, MACR, (MVK+MACR)/isoprene ratios, $O_3$, $NO_2$, temperature, CO and NO. The X-axis is "hour of day". The black thick line and red plus sign represent the median and mean value, respectively.





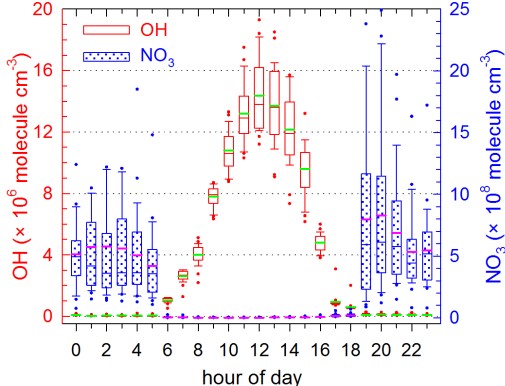

Fig. 4: Box and whisker plots of average diurnal patterns of modeled OH and NO₃ radical. The green and pink thick line represent the mean value.

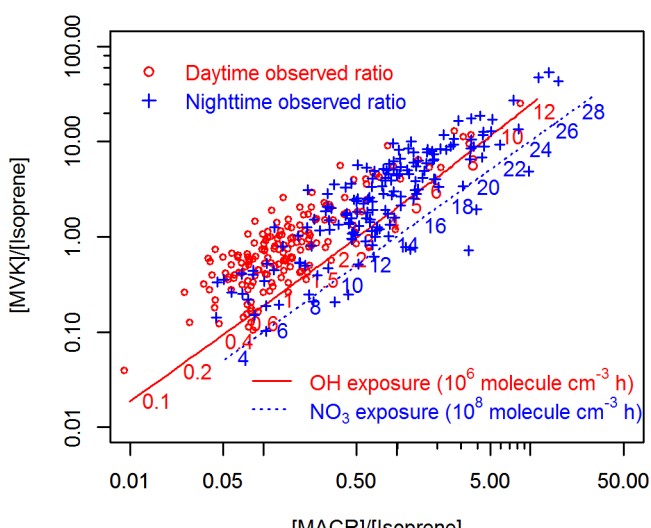

Fig. 5. Isoprene oxidation clock defined by the progression of daughter/parent (MVK/isoprene, MACR/isoprene) ratios (unit: molecules cm$^{-3}$ / molecules cm$^{-3}$). Red circles and blue crosses show the observed ratios for the daytime and nighttime measurements, respectively. The red solid and blue dashed lines are the results of isoprene sequential reaction scheme calculation. Texts next to the line indicate the theoretical exposures (the product of radical concentration and reaction time) corresponding to any given daughter–parent relationship.



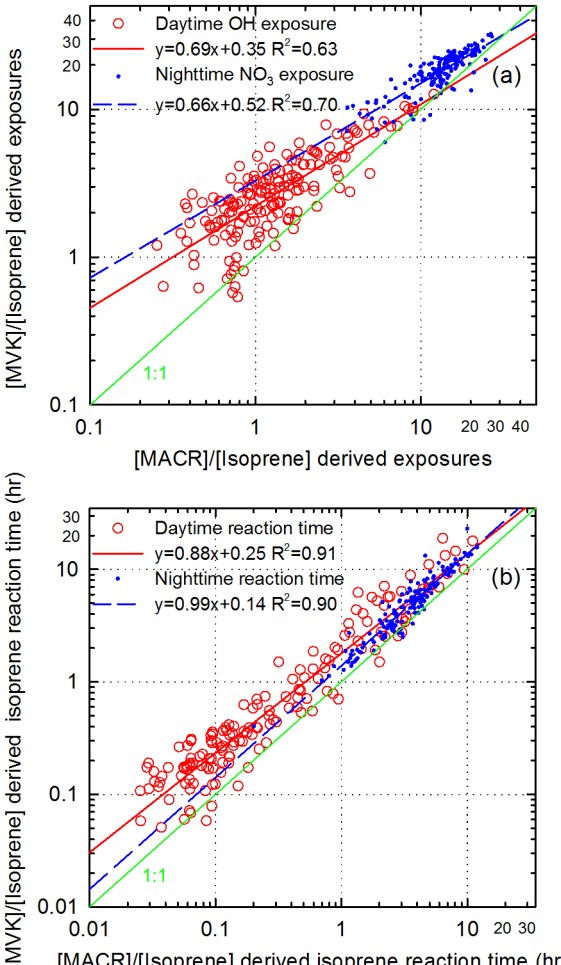

**Fig. 6: (a) Scatter plots of exposures derived from observed [MVK]/[isoprene] ratios versus that from [MACR]/[isoprene] ratios.**
**The unit of OH exposure and NO₃ exposure is $10^6$ molecules cm$^{-3}$ h and $10^8$ molecules cm$^{-3}$ h, respectively. The green line denotes a**
**1:1 relationship. (b) Isoprene reaction time derived from [MVK]/[isoprene] and [MACR]/[isoprene] method based on the modeled**
**OH and NO₃ concentrations. The green line denotes a 1:1 relationship.**



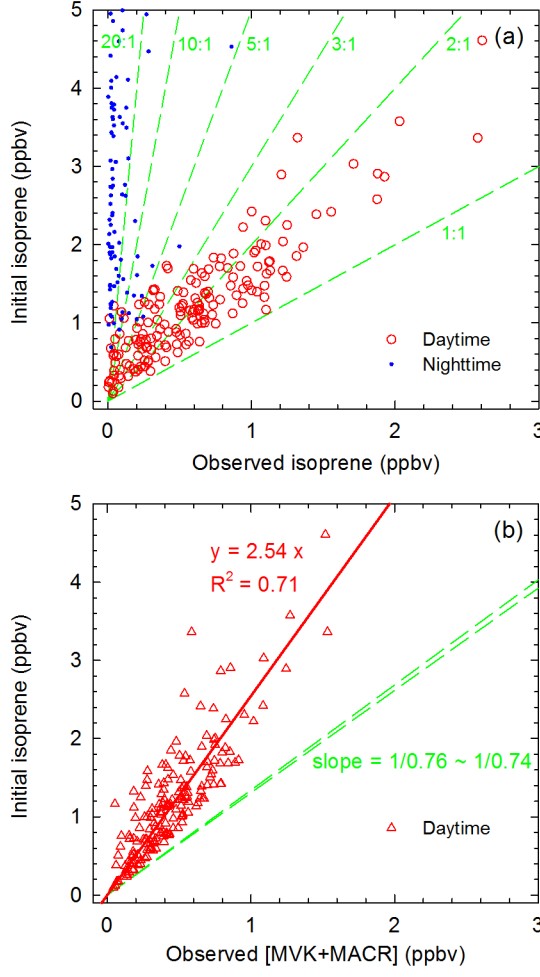

**Fig. 7: (a) Comparison of observed and initial isoprene mixing ratios. Green dashed lines denote slopes for different ratios of initial to observed isoprene. (b) Relationship between initial isoprene and measured [MVK+MACR] during the day. The green dashed lines denote slopes for different yields of (MVK+MACR) of the OH-initiated oxidation of isoprene for the ranges of the observed NO distribution (Fig. S1).**



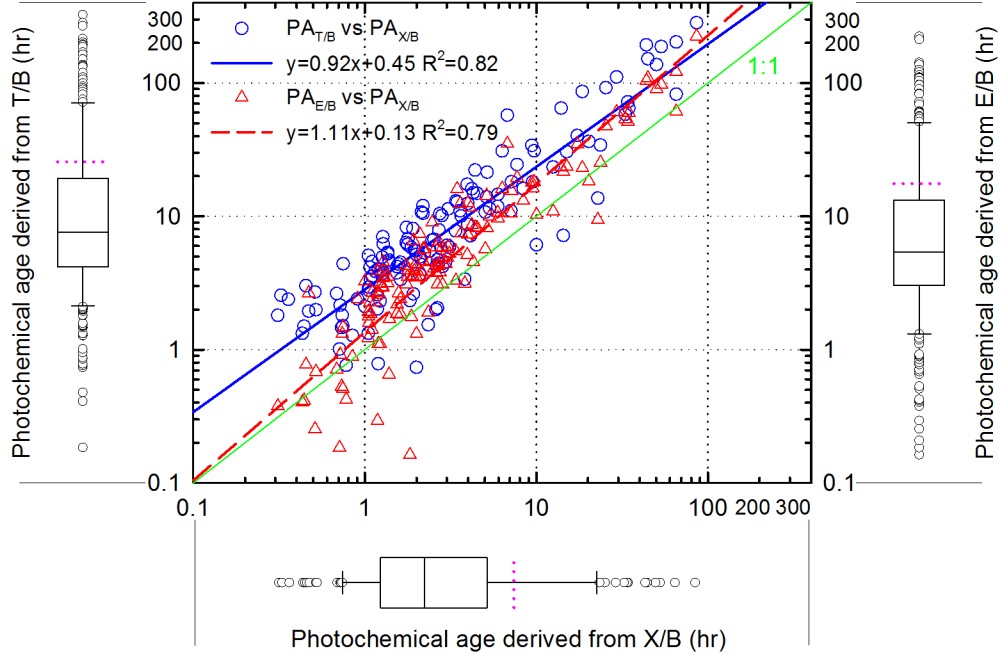

**Fig. 8: Photochemical age of the air mass derived from the daytime toluene/benzene (T/B), ethylbenzene/benzene (E/B) and m,p-xylene/benzene (X/B) ratios. The green line denotes a 1:1 relationship. Next to axes are the box and whisker plots of each result, and the pink dotted lines denote the mean values.**