# Peer review of "Low-level summertime isoprene observed at a forested mountaintop"

_Atmospheric Chemistry and Physics, 2018_

## Referee Comment (RC1) · Anonymous Referee #2 · 18 Jun 2018

The authors present field measurements of isoprene, MACR and MVK at a forested mountain site in southern China. Relatively lower concentration levels of isoprene and higher ratios of (MVK+MACR)/isoprene than other forest sites were observed. The authors argued that the low isoprene levels were ascribed to the strong atmospheric oxidative capacity in the region, and a chemical box model constrained with observations further confirmed this argument. Overall, this paper presents new data of isoprene and its first-stage degradation intermediates in a unique environment with interactions of both biogenic and anthropogenic emissions. It is well organized and written. Therefore,

this manuscript can be considered for publication after the following specific comments being properly addressed.

Major comments:

The authors used an MCM chemical box model to predict the concentrations of OH and NO3 radicals. This model has been applied in several previous studies. However, there are still several issues that need to be clarified, and some additional modeling runs are needed to check the sensitivity of modelling results to these issues.

-Some previous studies have suggested that the MCM model may not work well for reproducing the HOx concentrations at high-BVOCs and low-NOx conditions, which seems to be the case of study area in the present study. The authors may need to check the applicability of the MCM to the environmental condition at this forested mountain site.

-The NO2 measurement analyzer used in this study may significantly overestimate for NO2 at rural and remote sites. The authors are suggested to conduct more modeling analyses with artificially reduced NO2 concentrations to examine the sensitivity of predicted OH and NO3 to the input NO2 data.

-The heterogeneous reactions of N2O5 play an important role in the nocturnal NO3 chemistry. How does the PBM-MCM model present the N2O5 chemistry? What uptake coefficients of N2O5 onto particles were adopted in the model? Previous MCM modelling studies with addition of heterogeneous N2O5 chemistry have indicated the significant effect of this process on the simulation of photochemical processes (Xue et al., 2014). Additional modelling analyses are needed to examine the sensitivity of NO3 to the N2O5 heterogeneous chemistry.

-Similarly, the HONO chemistry was also not well represented in the MCM model, but plays an important role in the OH simulation (Xue et al., 2014). Did the PBM-MCM model take this chemistry into account? If not, the authors may need to conduct some

sensitivity analyses or at least mention the potential uncertainty of modeling results.

Xue et al., Ground-level ozone in four Chinese cities: precursors, regional transport and heterogeneous processes, Atmos. Chem. Phys., 14, 13175- 13188, 2014.

Sections 2.4 and 2.5: additional information about the calculation methods of the isoprene reaction time and air mass age, including principles and uncertainties, are required for readers to better understand and reproduce the results. The authors may also need to comment on the difference between reaction time and air mass age.

The major conclusion of this study is the strong regional atmospheric oxidative capacity leads to fast oxidation of isoprene in southern China. Some recent long-term observational studies have indicated the increasing trends of ozone concentrations in this region (Wang et al., 2009; Xue et al., 2014). As ozone is usually regarded as an indicator of the regional atmospheric oxidation capacity, these studies confirmed the increasing trend of atmospheric oxidizing capacity in southern China. It would be useful if the authors could discuss the trends of ozone and atmospheric oxidative capacity and comment on the projected trend in the future.

Wang et al., Increasing surface ozone concentrations in the background atmosphere of southern China, 1994-2007. Atmos. Chem. Phys., 9, 6217-6227, 2009.

Xue et al., Increasing external effects negate local efforts to control ozone air pollution: a case study of Hong Kong and implications for other Chinese cities, Environ. Sci. Tech., 48, 10769-10775, 2014.

Minor comments:

Page 1, Lines 22-23: this sentence is incomplete. Rephrase this sentence.

Page 2, Lines 27-31: the oxidation of isoprene by OH radicals is very complex. MACR and MVK can be further oxidized to form MGLY and other secondary compounds. Here it would be helpful to add several sentences to briefly summarize the thorough oxidation chemistry of isoprene as well as the major knowledge gaps in understanding

this chemistry.

Page 3, Lines 1-4: at nighttime, NO3 is generally in a thermal equilibrium with N2O5, which can be also taken up onto aerosols. Such heterogeneous reaction of N2O5 is an important sink of NOx at nighttime, and can compete with the reactions of NO3 with BVOCs. The authors are suggested to add several sentences to mention this process and provide a thorough picture of the nocturnal chemistry.

Page 3, Lines 20-21: rephrase this sentence.

Page 4, Line 19: Southeast Asia

Change "atmospheric boundary layer (APL)" to "planetary boundary layer (PBL)" throughout the manuscript.

Page 5, Lines 18-19: it has been confirmed that this commercial NOx analyzer with a default molybdenum oxide converter can significantly overestimate for NO2, especially in rural and remote areas such as the forested mountaintop in the present study. The authors may need clarify the detailed configuration of this NOx analyzer (e.g., the converters for converting NO2 to NO) and state the uncertainty of NO2 measurements if the MoO converter was used.

Xu et al., Evaluating the uncertainties of thermal catalytic conversion in measuring atmospheric nitrogen dioxide at four differently polluted sites in China, Atmos. Environ., 76, 221-226, 2013.

Page 7, Lines 7-9: it is a little bit strange that the observed concentrations of benzene (and also toluene) are very low. The lifetime of benzene is rather long and thus its ambient abundances are usually not so low.

Page 7, Lines 9-11: provide standard deviations for the averages.

Page 8, Lines 13-14: rephrase this sentence "Although . . ."

Page 8, Line 20: on the source of MVK and MACR at this site, I wonder if regional

transport could also contribute to the observed MVK and MACR. What are the lifetimes of MVK and MACR?

Page 10, Line 17, "the observed relationship of observed MVK/isoprene": delete one "observed"

Table 1: provide the standard deviation of the average concentrations if available.

[Figure]

---

## Short Comment (SC1) · 7 Aug 2018

*Corresponding to: tbongue@jnu.edu.cn*

Dear Referee#2,

We greatly appreciate the time and effort that you spent in reviewing our manuscript. The comments are really thoughtful and helpful to improve the quality of our paper. Most of the modifications were made in the manuscript (attached) and below are point-by-point response to these comments. There follows a list of referee comments (**in black**), together with our response (**in blue**) and details of corrections/improvements to the manuscript (**in red**).
* * *
The authors present field measurements of isoprene, MACR and MVK at a forested mountain site in southern China. Relatively lower concentration levels of isoprene and higher ratios of (MVK+MACR)/isoprene than other forest sites were observed. The authors argued that the low isoprene levels were ascribed to the strong atmospheric oxidative capacity in the region, and a chemical box model constrained with observations further confirmed this argument. Overall, this paper presents new data of isoprene and its first-stage degradation intermediates in a unique environment with interactions of both biogenic and anthropogenic emissions. It is well organized and written. Therefore, this

manuscript can be considered for publication after the following specific comments being properly addressed.

Response: We thank the referee for this overview. We have made changes to the manuscript based on the referee's helpful comments.
* * *
The authors used an MCM chemical box model to predict the concentrations of OH and $NO_3$ radicals. This model has been applied in several previous studies. However, there are still several issues that need to be clarified, and some additional modeling runs are needed to check the sensitivity of modelling results to these issues.

-**1.** Some previous studies have suggested that the MCM model may not work well for reproducing the $HO_x$ concentrations at high-BVOCs and low-$NO_x$ conditions, which seems to be the case of the study area in the present study. The authors may need to check the applicability of the MCM to the environmental condition at this forested mountain site.

Response: Many thanks and we agree with the referee that MCM needs to be improved and further optimized at pristine forest environments. And also as the OH mixing ratios modelled by PBM-MCM can only represent the levels at the site. Therefore, a parameterization method using measured aromatic hydrocarbons has been applied to estimate the regional mixing ratios of daytime OH radicals (Fig.1 and Fig.2) (Shiu et al., 2007). We think that this additional approach provides a good complement to the current model for evaluating the atmospheric oxidative capacity in the present forest regions. We have therefore added this point in the Section 2.3 and 3.3.1.

In the Section 2.3:" Several issues should be noted in applying PBM-MCM to the present study. The first is that some previous studies have suggested that MCM may not work well for reproducing the OH concentrations in pristine forest environments (Kim et al., 2015). And also as the OH mixing ratios modelled by PBM-MCM can only represent the levels at the site. To provide a complement to the PBM-MCM model, a parameterization method using measured aromatic hydrocarbons was applied to

estimate the regional mixing ratios of daytime OH radicals (Shiu et al., 2007). Details about the parameterization method are given in the Text S1."

In the Section 3.3.1:"…The regional mixing ratios of OH ($19.7 \pm 2.3 \times 10^6$ molecules cm$^{-3}$) were even higher than the modelled site-level OH ($11.7 \pm 0.4 \times 10^6$ molecules cm$^{-3}$) during 09:00 − 15:00 LST (Fig. S1), though the results at 11:00 LST estimated by the two methods were comparable ($13.2 \pm 0.7$ and $13.9 \pm 4.3 \times 10^6$ molecules cm$^{-3}$) (Fig. S2). Overall,…".

[Figure]

Fig.1. Scatterplots of the regional mixing ratios of OH during 09:00 − 15:00 LST derived from the toluene/benzene (OH$_{T/B}$), ethylbenzene/benzene (OH$_{E/B}$) and m,p-xylene/benzene (OH$_{X/B}$) ratios. The green line denotes a 1:1 relationship. Next to axes are the box and whisker plots of each result, and the pink dotted lines denote the mean values.

[Figure]

Fig.2. Hourly variations of OH concentrations derived from the toluene/benzene ratio (OH$_{T/B}$), ethylbenzene/benzene ratio (OH$_{E/B}$), m,p-xylene/benzene ratio (OH$_{X/B}$) and PBM-MCM during 09:00 − 15:00 LST. Short-dashed lines denote the mean values of each method. Error bars indicate the 95% confidence interval.

*References:*

*Shiu, C. J., Liu, S. C., Chang, C. C., Chen, J. P., Chou, C. C. K., Lin, C. Y., and Young, C. Y.: Photochemical production of ozone and control strategy for Southern Taiwan, Atmos Environ, 41, 9324-9340, 10.1016/j.atmosenv.2007.09.014, 2007.*

*Kim, S., Kim, S. Y., Lee, M., Shim, H., Wolfe, G. M., Guenther, A. B., He, A., Hong, Y., and Han, J.: Impact of isoprene and HONO chemistry on ozone and OVOC formation in a semirural South Korean forest, Atmos Chem Phys, 15, 4357-4371, 10.5194/acp-15-4357-2015, 2015.*
* * *
-**2.** The $NO_2$ measurement analyzer used in this study may significantly overestimate for $NO_2$ at rural and remote sites. The authors are suggested to conduct more modeling analyses with artificially reduced $NO_2$ concentrations to examine the sensitivity of predicted OH and $NO_3$ to the input $NO_2$ data.

Response: Many thanks for spotting this, we have therefore conducted sensitivity analyses of modelled daytime OH and nighttime $NO_3$ with cutting $NO_2$ concentrations. The OH and $NO_3$ concentrations showed decreasing trends with the $NO_2$ reducing (Fig.3). Assuming the average concentrations of daytime and nighttime $NO_2$ have been overestimated by 64.4 ± 2.9% and 62.4 ± 3.0% (Fig. 4), respectively, according to the results at a high-altitude mountain site in China (Xu et al., 2013), the mean daytime OH concentration at the present site would decrease from $7.3 ± 0.5 × 10^6$ molecules $cm^{-3}$ to $5.5 ± 0.4 × 10^6$ molecules $cm^{-3}$, with a reduction rate of 23.8 ± 6.2%. And for nighttime $NO_3$, the decreasing rate was 41.5 ± 5.2% (from $6.0 ± 0.5 × 10^8$ molecules $cm^{-3}$ to $3.5 ± 0.3 × 10^8$ molecules $cm^{-3}$). We have therefore added discussions about this point in the Section 2.3 and 3.3.3.

In the Section 2.3: "…Second, the $NO_2$ concentrations, an important input into PBM-MCM, may be significantly overestimated at this remote mountaintop site that receives a considerable amount of photochemically aged air (Xu et al., 2013). To examine the sensitivity of predicted OH and $NO_3$ to the input $NO_2$ data, we have therefore conducted several modelling analyses with artificially reduced $NO_2$ concentrations. More details are discussed in Section 3.3.3…."

In the Section 3.3.3: "As mentioned in Section 2.3, the overestimated $NO_2$ and unmeasured HONO were the two main uncertainties in applying the PBM-MCM to evaluate the atmospheric oxidative capacity in the present study. As for the first issue, we conducted sensitivity analysis of modelled OH and $NO_3$ with artificially reduced $NO_2$ concentrations for the period Aug. 11 − Aug. 15 2016. The OH and $NO_3$ concentrations showed decreasing trends with the $NO_2$ cutting (Fig. S9). Assuming the average

concentrations of daytime and nighttime $NO_2$ have been overestimated by 64.4 ± 2.9% and 62.4 ± 3.0%, respectively, according to the results at a high-altitude mountain site in China (Xu et al., 2013), the mean daytime OH concentration would decrease from $7.3 ± 0.5 × 10^6$ molecules $cm^{-3}$ to $5.5 ± 0.4 × 10^6$ molecules $cm^{-3}$, with a reduction rate of 23.8 ± 6.2%. And for nighttime $NO_3$, the decreasing rate was 41.5 ± 5.2% (from $6.0 ± 0.5 × 10^8$ molecules $cm^{-3}$ to $3.5 ± 0.3 × 10^8$ molecules $cm^{-3}$)….".

[Figure]

Fig.3. Sensitivity analysis of PBM-MCM modelled daytime OH and nighttime $NO_3$ with reduced $NO_2$ concentrations for the period Aug.11 – Aug.15 2016.

[Figure]

Fig.4. Hourly modification factor of $NO_2$ during July 15–August 17 2016 at the Nanling site. The data in the figure are reproduced from the study conducted at a high-altitude mountain site (Mt. Tai) in central China by Xu et al. (2013).

In the Section 3.3.3: "…Another issue was the lack of HONO measurements, we have therefore conducted two days' sensitivity analyses (Aug. 13 and Aug. 15 2016) by using the average diurnal profiles of HONO observed at a background site (Hok Tsui) in Hong Kong in autumn 2012 (Zha, 2015). The results showed that daytime OH concentrations with HONO taken into account was 22 ± 19% higher than that without HONO (Fig. S10). Nonetheless, the results are not representative since the HONO mixing ratios were not observed in the present study. Thus, certain uncertainties remain in the present modelling for HONO chemistry….Overall, we made an attempt to took these uncertainties into account by assuming the modification factor based on the above sensitivity analyses (Fig. 4). On average, current PBM-MCM may have 19 ± 9% and 73 ± 16% overestimation of the daytime OH and nighttime $NO_3$ concentrations, respectively.".

[Figure]

Fig.5. Sensitivity analysis of PBM-MCM modelled daytime OH with and without HONO for the period Aug.13 and Aug.15 2016. The HONO data was obtained from the study conducted at a background site (Hok Tsui) in Hong Kong in autumn 2012 by Zha (2015).

In the Section 2.4, uncertainties added: "Two potential uncertainties arise when applying this approach to the low-NO atmosphere in the present study (Wolfe et al., 2016). First, the yields of MVK and MACR from OH-initiated isoprene oxidation are a nonlinear function of NO. Previous applications of this method (de Gouw, 2005;Roberts et al., 2006;Stroud et al., 2001;Karl et al., 2007;Kuhn et al., 2007) have assumed lab-derived constant yields for MVK and MACR, respectively (Atkinson and Arey, 2003), which may not be appropriate in the present case. In this study, we chose the results in Master Chemical Mechanism (MCM) v3.3.1 (Jenkin et al., 2015). Second, the reaction time implied by any observed daughter/parent ratio depends on the concentration of radical (*i.e.*, OH and $NO_3$), which was not measured and varies as an air mass ages. Rather than assume a single "typical" value for radical, we express reaction time in terms of "exposure", defined here as the product of radical concentration and reaction time. Also, the atmospheric reaction time of isoprene can be calculated based on the modeled OH and $NO_3$ concentrations in this study."

In the Section 2.5, principles and uncertainties added: "The limitations of using aromatic ratios to calculate photochemical age are addressed by previous studies (Parrish et al., 2007). Three issues should be noted when applying this method to present study. The first is the mixing of fresh emissions with aged air masses caused by the transport processes in the atmosphere. Second, the modelled OH at this site can't represent the regional value in the air parcel. And third, the effect of horizontal and vertical mixing are similar for two compounds of BTEX ratio. Here we assume that there were no fresh emissions of those aromatics along the paths travelled and the OH concentrations were constant during hourly modelling duration throughout the pristine forest atmosphere. Regardless of those uncertainties, this air mass age estimation method still provides useful indicators of the degree of photochemical processing in the atmosphere."

As for the differences between reaction time and air mass age, we have added an explanation of reaction time and a description of air mass age in the Section 2.4 and Section 2.5, respectively.

In the Section 2.4: "…the time of isoprene in the atmosphere between emission and detection,…"

In the Section 2.5: "The estimation of isoprene reaction time in an air parcel provides useful information on the isoprene oxidation rates. In addition, measurements of certain anthropogenic VOCs provided us a chance to evaluate the aging degree of the air mass via the determination of photochemical age which

can be calculated by the ratios of two VOC species that share common emission sources but with large different reactivities with OH (Parrish et al., 2007). It is well known that BTEX (benzene, toluene, ethylbenzene, and xylenes) provide particularly promising VOC ratios for following photochemical processing on timescales of hours to days (de Gouw, 2005;Shiu et al., 2007;Yuan et al., 2012;Yang et al., 2017)."

Response: Many thanks for spotting this, corrections have been made to read "This paper is structured as follows. Firstly, an overview of the measured concentrations and diurnal variations of isoprene and

its oxidation products were given. Then the calculated levels of daytime OH and nighttime $NO_3$ were presented and discussed. And furthermore, we describe in detail the atmospheric reaction time of isoprene, along with brief estimations of initial isoprene and the air mass age. Finally, concluding remarks including a synthesis of current findings and some implications are presented."
* * *
-**11.** Page 4, Line 19: Southeast Asia

Response: Many thanks for spotting this – correction made.
* * *
-**12.** Change "atmospheric boundary layer (ABL)" to "planetary boundary layer (PBL)" throughout the manuscript.

Response: Many thanks, correction made.
* * *
-**13.** Page 5, Lines 18-19: it has been confirmed that this commercial $NO_x$ analyzer with a default molybdenum oxide converter can significantly overestimate for $NO_2$, especially in rural and remote areas such as the forested mountaintop in the present study. The authors may need clarify the detailed configuration of this $NO_x$ analyzer (e.g., the converters for converting $NO_2$ to NO) and state the uncertainty of $NO_2$ measurements if the MoO converter was used.

*Xu et al., Evaluating the uncertainties of thermal catalytic conversion in measuring atmospheric nitrogen dioxide at four differently polluted sites in China, Atmos. Environ., 76, 221-226, 2013.*

Response: We agree with the referee on the impact of molybdenum oxide converter may have on the measured $NO_2$ concentrations and we have added a discussion accordingly in the Section 2.2.2 and Section 3.1, respectively.

In the Section 2.2.2: "Oxides of nitrogen ($NO$-$NO_2$-$NO_x$) were measured at 1 min resolution using chemiluminescence analyser (Model 42i-TL, Thermo Scie ntific, Inc.), which has a detection limit of 50 pptv. $NO_2$ is converted to NO by a heated molybdenum converter before it can be measured by the

chemiluminescence detection of NO. Studies have shown that the converter may overestimate $NO_2$ as the measured $NO_2$ may have included other oxidized nitrogen compounds (Xu et al., 2013), and thus, the $NO_2$ concentrations given below are considered the upper limits of their actual values."

In the Section 3.1: "It should be noted that the measured $NO_2$ concentrations are in fact upper limits, because the $NO_2$ probably includes some oxidized reactive nitrogen that was converted by the molybdenum. It has been confirmed that the $NO_2$ concentrations measured by analyzer with internal molybdenum oxide converter can be significantly overestimated in areas far away from fresh $NO_x$ emission sources such as the remote mountaintop in the present study (Xu et al., 2013). Therefore, we made a modifying to the observed $NO_2$ by adopting the modification factors (Fig. S8) obtained at Mountain Tai, a high-altitude site (1533 m a.s.l.) in central-eastern China (Xu et al., 2013). The modified $NO_2$ (889 ± 27 pptv) was 1.1−2.5 times (1.8 ± 0.3) lower than that observed."
* * *
14. Page 7, Lines 7-9: it is a little bit strange that the observed concentrations of benzene (and also toluene) are very low. The lifetime of benzene is rather long and thus its ambient abundances are usually not so low.

Response: Thanks for your good question. We are also surprised by the low-level benzene and toluene, and the possible reasons may include:

(1) Long distance of the site away from city centers. The site is located in the free troposphere (FT) or upper planetary boundary layer (PBL) of southern China. Therefore, the low observed values suggest that the diffusion of benzene and toluene from ground source emissions within the PBL to the FT is strong, and thus they might have reached a steady state in this remote and high-elevation atmosphere.

(2) The strong oxidizing power of the troposphere in PRD region. Intensive studies have pointed to worsening photochemical pollution and strong atmospheric oxidative capacity throughout this region. Thus the chemical removal of benzene and toluene by radicals (e.g., OH) occurs rapidly and thoroughly along the paths air parcels traveled in the absence of mixing with fresh emissions in this pristine forest region.

(3) The high relative humidity (92.2 ± 8.3%) and cloudy conditions are favorable to the removal of benzene and toluene at this site, which may also lower the levels to a certain extent.

We have therefore added a discussion accordingly in the Section 3.1: "…Aromatic compounds showed very low mixing ratios, among which, toluene is the most abundant (154 ± 20 pptv, average ± 95% confidence interval, the same below), followed by benzene (51 ± 8 pptv), ethylbenzene (47 ± 6 pptv) and m,p-xylene (38 ± 4 pptv). Being remote from major anthropogenic emission sources, the ambient aromatics levels at this site were significantly low compared to the abundances that measured at a regional background site in the PRD region, while they were comparable with other remote mountain sites. The low-level aromatics provide additional evidences showing that the site is remote and suffering from minor influence of local anthropogenic emissions."
* * *
-**15.** Page 7, Lines 9-11: provide standard deviations for the averages.

Response: Many thanks, done.
* * *
-**16.** Page 8, Lines 13-14: rephrase this sentence "Although …"

Response: Many thanks for spotting this, correction made.
* * *
-**17.** Page 8, Line 20: on the source of MVK and MACR at this site, I wonder if regional transport could also contribute to the observed MVK and MACR. What are the lifetimes of MVK and MACR?

Response: We appreciate the referee's question and the possible regional transport of MVK and MACR are explained in the following.

The lifetimes of MVK and MACR by reaction with OH are 1.9 and 1.0 hours, respectively, assuming 12-h daytime OH = $8.0 \times 10^6$ molecules cm$^{-3}$. The average daytime wind speed during the sampling periods was 3.9 $\pm$ 0.2 m s$^{-1}$ at the site, and the distance between the sampling site and the nearest urban

center is 38 km (Fig. 6), the air parcel from upwind locations would spend about 2.7 hours to arrive at the sampling site. This is enough time for the depletion of MVK and MACR along the traveling path during the daytime.

However, the nighttime chemical oxidation of MVK and MACR was slow, with lifetimes of MVK and MACR by $NO_3$ oxidation of 0.5 years and 72 hours, respectively, assuming 12-h nighttime $NO_3$ = 5.0 × $10^8$ molecules $cm^{-3}$. Apart from biogenic sources, the anthropogenic sources of MVK and MACR, e.g., motor vehicles, biomass burning and industrial sources, have been reported by many studies (Borbon et al., 2001;Wagner and Kuttler, 2014;Hsieh et al., 2016;Diao et al., 2016). Therefore, the regional transport of MVK- or MACR- laden air could affect the observed nighttime levels at the site.

The "polarPlot" technique was therefore used for source identification (Fig. 7). The red dotted sectorial domains in the figure are interpreted as the regional transport interference as the concentrations of species increase with increasing wind speed. The high levels of species at high wind speeds most likely came from the nearby urban centers. Therefore, measurements that are deemed to be affected by regional transport are all excluded from the dataset in the analysis.

We have therefore added a discussion accordingly in Section 3.2: "…Moreover, MVK and MACR could be also transported from anthropogenic sources in neighbour cities to the sampling site at night (More detailed discussion is given in Text S5)…."

[Figure]

Fig. 6. (a) Location of the Nanling site, Dinghu Mountain site, Hok Tsui site, Guangzhou and Hong Kong. The Nanling site is 174 km northeast to the Dinghu Mountain site. Red outlined domain represent the Pearl River Delta region. (b) Map showing the nearest urban centers (Yangshan County, Ruyuan County, Lechang City, Lianzhou City, Shaoguan City and Yingde City) around the site.

[Figure]

Fig. 7. Daytime and nighttime polarplots of MVK and MACR during the sampling period (July 15–August 17 2016). Concentrations varied by wind speed (ws, unit in m/s) and wind direction. Red dotted sectorial domains represent the interferences of regional transport from nearby urban centers.

* * *
**-18.** Page 10, Line 17, "the observed relationship of observed MVK/isoprene": delete one "observed"

Response: Many thanks, correction made.
* * *
**-19.** Table 1: provide the standard deviation of the average concentrations if available.

Response: Many thanks, done.

---

## Referee Comment (RC2) · Anonymous Referee #3 · 9 Aug 2018

Gong et al. presents their results of online observations of isoprene and its first-stage oxidation products MVK and MACR in summer 2016 at a remote, high-altitude mountain forest site to the north of the air-polluted PRD region in southern China. They found that the isoprene level was significantly lower and attributed it to the strong regional atmospheric oxidative capacity. The PBM-MCM model was used to estimate the OH and NO3 concentrations to support their assumptions. The paper is well written and organized. The reviewer would recommend the manuscript for publication after some specific comments. Specific Comments: 1. O3, OHx, PAN, and NO3 are indi-

cators of atmospheric oxidative capacity. Since OH and NO3 were not determined in the observation, the observed O3 concentration is a more powerful tool to express the atmospheric oxidative capacity. The diurnal variations of O3 peaked at 20:00 is very interesting, because the changing trends of O3 and sun radiation were not accordant. The temporal variations of O3 also show different trends during the observation. Could regional transport contribute O3 to the measurement site? The authors had better add more discussion on the variations of O3 concentration. 2. The modelled OH and NO3 concentrations were regarded as the most important evidence for the conclusion of this manuscript. However, the PBM-MCM model is not a good tool to estimate OH concentrations at low NOx concentrations at remote site like this study. The reviewer strongly recommend the authors add some other models to support their conclusions. 3. Page 1, Line 22-23, this sentence is incomplete. 4. Page 4, Line 32, the specifications of the Teflon filter should be clarified. 5. Page 5, Line 21-23, it is confused that "daily" and "every two days". Secondly, it seems that SO2, NOx, and CO analyzers are usually calibrated with domestic standard gases which are not NIST-traceable. The NIST-traceable standard was only applied to calibrate O3 analyzer.

---

## Author Comment (AC1) · 18 Aug 2018

*Corresponding to: tbongue@jnu.edu.cn*

Dear Referee #3,

We greatly appreciate the time and effort that you spent in reviewing our manuscript. The comments are really thoughtful and helpful to improve the quality of our paper. Most of the modifications were made in the manuscript and below are point-by-point response to these comments. There follows a list of referee comments (**in black**), together with our response (**in blue**) and details of corrections/improvements to the manuscript (**in red**).
* * *
Gong et al. presents their results of online observations of isoprene and its first-stage oxidation products MVK and MACR in summer 2016 at a remote, high-altitude mountain forest site to the north of the air-polluted PRD region in southern China. They found that the isoprene level was significantly lower and attributed it to the strong regional atmospheric oxidative capacity. The PBM-MCM model was used to estimate the OH and $NO_3$ concentrations to support their assumptions. The paper is well written and organized. The reviewer would recommend the manuscript for publication after some specific comments.

Response: We thank the referee for this overview. We have made changes to the manuscript based on the referee's helpful comments.
* * *
Specific Comments:

**-1.** $O_3$, $OH_x$, PAN, and $NO_3$ are indicators of atmospheric oxidative capacity. Since OH and $NO_3$ were not determined in the observation, the observed $O_3$ concentration is a more powerful tool to express the atmospheric oxidative capacity. The diurnal variations of $O_3$ peaked at 20:00 is very interesting, because the changing trends of $O_3$ and sun radiation were not accordant. The temporal variations of $O_3$ also show different trends during the observation. Could regional transport contribute $O_3$ to the measurement site? The authors had better add more discussion on the variations of $O_3$ concentration.

Response: Many thanks for spotting this. We noticed that the high levels of $O_3$, as well as its distinct diurnal variations, were observed at the site, and we are preparing another paper to discuss these points in details. In this revised manuscript, discussions about the high abundances of $O_3$ at the site has been added in the Section 3.1. As for the interesting $O_3$ diurnal variations, a number of studies have observed a similar pattern at mountaintops, that is, high concentrations at night and low concentrations during the daytime (Gallardo et al., 2000;Gao et al., 2017). Some previous studies have concluded that the factors closely related to the distinct $O_3$ diurnal patterns at mountaintops were the boundary layer diurnal cycles (Gao et al., 2017), the mountain-valley breezes (Zaveri et al., 1995;Yang et al., 2012;Cristofanelli et al., 2013), the regional transport effects (Naja et al., 2003;Li et al., 2008;Zhang et al., 2015;Gao et al., 2017), the location and altitude of a mountain (Chevalier et al., 2007;Monteiro et al., 2012), and ozone vertical distributions (Forrer et al., 2000;Zellweger et al., 2003;Gheusi et al., 2011). Overall, the distinct diurnal variations in $O_3$ concentrations are the result of a combination of various physical and chemical processes. This point has been added to the Section 3.2.

In the Section 3.1:" High abundances of $O_3$ at this site likely indicate strong oxidizing power of the present remote atmospheres."

In the Section 3.2:" Some previous studies at mountaintops observed distinct diurnal variations in $O_3$ concentrations that featured with high levels at night and low levels during the daytime (Zaveri et al.,

1995;Gao et al., 2017), which are the result of a combination of various physical and chemical processes (e.g., boundary layer diurnal cycles, mountain-valley breezes, regional transport, photochemical reactions)."

*References:*

*Zaveri, R. A., Saylor, R. D., Peters, L. K., McNider, R., and SonG, A.: A model investigation of summertime diurnal ozone behavior in rural mountainous locations, Atmos Environ, 29, 1043-1065, 1995.*

*Forrer, J., Ruttimann, R., Schneiter, D., Fischer, A., Buchmann, B., and Hofer, P.: Variability of trace gases at the high-Alpine site Jungfraujoch caused by meteorological transport processes, Journal of Geophysical Research-Atmospheres, 105, 12241-12251, Doi 10.1029/1999jd901178, 2000.*

*Gallardo, L., Carrasco, J., and Olivares, G.: An analysis of ozone measurements at Cerro Tololo (30 degrees S, 70 degrees W, 2200 m.a.s.l.) in Chile, Tellus Series B-Chemical and Physical Meteorology, 52, 50-59, DOI 10.1034/j.1600-0889.2000.00959.x, 2000.*

*Naja, M., Lal, S., and Chand, D.: Diurnal and seasonal variabilities in surface ozone at a high altitude site Mt Abu (24.6 degrees N, 72.7 degrees E, 1680 m asl) in India, Atmos Environ, 37, 4205-4215, 10.1016/s1352-2310(03)00565-x, 2003.*

*Zellweger, C., Forrer, J., Hofer, P., Nyeki, S., Schwarzenbach, B., Weingartner, E., Ammann, M., and Baltensperger, U.: Partitioning of reactive nitrogen ($NO_y$) and dependence on meteorological conditions in the lower free troposphere, Atmos Chem Phys, 3, 779-796, 2003.*

*Chevalier, A., Gheusi, F., Delmas, R., Ordonez, C., Sarrat, C., Zbinden, R., Thouret, V., Athier, G., and Cousin, J. M.: Influence of altitude on ozone levels and variability in the lower troposphere: a ground-based study for western Europe over the period 2001-2004, Atmos Chem Phys, 7, 4311-4326, DOI 10.5194/acp-7-4311-2007, 2007.*

*Li, J., Pochanart, P., Wang, Z. F., Liu, Y., Yamaji, K., Takigawa, M., Kanaya, Y., and Akimoto, H.: Impact of Chemical Production and Transport on Summertime Diurnal Ozone Behavior at a Mountainous Site in North China Plain, Sola, 4, 121-124, 10.2151/sola.2008-031, 2008.*

*Gheusi, F., Ravetta, F., Delbarre, H., Tsamalis, C., Chevalier-Rosso, A., Leroy, C., Augustin, P., Delmas, R., Ancellet, G., Athier, G., Bouchou, P., Campistron, B., Cousin, J. M., Fourmentin, M., and Meyerfeld, Y.: Pic 2005, a field campaign to investigate low-tropospheric ozone variability in the Pyrenees, Atmos Res, 101, 640-665, 10.1016/j.atmosres.2011.04.014, 2011.*

*Monteiro, A., Strunk, A., Carvalho, A., Tchepel, O., Miranda, A. I., Borrego, C., Saavedra, S., Rodriguez, A., Souto, J., Casares, J., Friese, E., and Elbern, H.: Investigating a high ozone episode in a rural mountain site, Environmental pollution, 162, 176-189, 10.1016/j.envpol.2011.11.008, 2012.*

*Yang, C. F. O., Lin, N. H., Sheu, G. R., Lee, C. T., and Wang, J. L.: Seasonal and diurnal variations of ozone at a high-altitude mountain baseline station in East Asia, Atmos Environ, 46, 279-288, 10.1016/j.atmosenv.2011.09.060, 2012.*

*Cristofanelli, P., di Carlo, P., Altorio, A., Dari Salisburgo, C., Tuccella, P., Biancofiore, F., Stocchi, P., Verza, G. P., Landi, T. C., Marinoni, A., Calzolari, F., Duchi, R., and Bonasoni, P.: Analysis of Summer Ozone Observations at a High Mountain Site in Central Italy (Campo Imperatore, 2388 m a.s.l.), Pure Appl. Geophys., 170, 1985-1999, 10.1007/s00024-012-0630-1, 2013.*

*Zhang, L., Jin, L. J., Zhao, T. L., Yin, Y., Zhu, B., Shan, Y. P., Guo, X. M., Tan, C. H., Gao, J. H., and Wang, H. L.: Diurnal variation of surface ozone in mountainous areas: Case study of Mt. Huang, East China, Science of the Total Environment, 538, 583-590, 10.1016/j.scitotenv.2015.08.096, 2015.*

*Gao, J., Zhu, B., Xiao, H., Kang, H., Hou, X., Yin, Y., Zhang, L., and Miao, Q.: Diurnal variations and source apportionment of ozone at the summit of Mount Huang, a rural site in Eastern China, Environmental pollution, 222, 513-522, 10.1016/j.envpol.2016.11.031, 2017.*
* * *
**-2.** The modelled OH and NO₃ concentrations were regarded as the most important evidence for the conclusion of this manuscript. However, the PBM-MCM model is not a good tool to estimate OH concentrations at low NOₓ concentrations at remote site like this study. The reviewer strongly recommend the authors add some other models to support their conclusions.

Response: Thanks for this comment. We agree with the referee that PBM-MCM indeed needs improvement and further optimization for its application under low-NOₓ environments.

Thus on one hand we conducted sensitivity analyses of the PBM-MCM modelled OH concentrations with certain uncertainties. The results showed that the current PBM-MCM may have 19 ± 9% overestimation of the daytime OH mixing ratios at the present study. See also our response to Referee #2, comment #2 and #4 for more detailed discussion on this point.

On the other hand, a widely-used parameterization method with measured aromatic hydrocarbons has been applied to estimate the regional mixing ratios of daytime OH radicals (Shiu et al., 2007). This additional approach can provide a good complement to the current model for evaluating the atmospheric oxidative capacity in the present forest regions. We have therefore added this point in the revised manuscript and supplement. See also our response to Referee #2, comment #1 for more detailed discussion of this point.

We sincerely appreciate the suggestions made by the Referees, and agree that more observations and modeling studies will be needed to address this question. We will consider the application of other models as a better diagnostic tool in the future, e.g., OBM-AOCP (Observation-Based Model for investigating Atmospheric Oxidative Capacity and Photochemistry) developed by Xue et al. (2016).

*References:*
*Shiu, C. J., Liu, S. C., Chang, C. C., Chen, J. P., Chou, C. C. K., Lin, C. Y., and Young, C. Y.: Photochemical production of ozone and control strategy for Southern Taiwan, Atmos Environ, 41, 9324-9340, 10.1016/j.atmosenv.2007.09.014, 2007.*
*Xue, L. K., Gu, R. R., Wang, T., Wang, X. F., Saunders, S., Blake, D., Louie, P. K. K., Luk, C. W. Y., Simpson, I., Xu, Z., Wang, Z., Gao, Y., Lee, S. C., Mellouki, A., and Wang, W. X.: Oxidative capacity and radical*

*chemistry in the polluted atmosphere of Hong Kong and Pearl River Delta region: analysis of a severe photochemical smog episode, Atmos Chem Phys, 16, 9891-9903, 10.5194/acp-16-9891-2016, 2016.*
* * *
**-3.** Page 1, Line 22-23, this sentence is incomplete.

Response: Many thanks for spotting this, the correction has been made to read "To investigate the atmospheric oxidizing capacity in forested high mountain areas adjacent to the photochemistry-active Pearl River Delta (PRD) region in southern China, one-month online observations of isoprene and its first-stage oxidation products methyl vinyl ketone (MVK) and methacrolein (MACR) were conducted at a national background station in summer 2016.".
* * *
**-4.** Page 4, Line 32, the specifications of the Teflon filter should be clarified.

Response: Many thanks for spotting this, corrections have been made to read "…a Teflon filter (0.25 µm pore size, 47 mm OD, Millipore, USA) was placed in front of the sample inlet.".
* * *
**-5.** Page 5, Line 21-23, it is confused that "daily" and "every two days". Secondly, it seems that $SO_2$, $NO_x$, and CO analyzers are usually calibrated with domestic standard gases which are not NIST-traceable. The NIST-traceable standard was only applied to calibrate $O_3$ analyzer.

Response: Many thanks for spotting this mistake, corrections have been made to read "All instruments were calibrated weekly by using a multi-gas calibrator (Model 146i, Thermo Scientific, Inc.) throughout the study. The NIST-traceable (National Institute of Standards and Technology, USA) standard was applied to calibrate the $O_3$ analyser. For $NO_x$, $SO_2$ and CO, standard gases provided by NRCCRM (National Research Center for Certified Reference Materials, China) was applied. Zero and span checks of all analysers were performed every two days.".

---

## Author Response (AR2)

*Corresponding to: tbongue@jnu.edu.cn*

We greatly appreciate the time and effort that the Referees spent in reviewing our manuscript. The comments are really thoughtful and helpful to improve the quality of our paper. We have addressed each comment below, with the Referee comment in **black** text, our response in **blue** text, and relevant manuscript changes noted in **red** text. In addition, the readability of this manuscript has been improved by slightly changing the structure and polishing the language. The revised manuscript with relevant changes marked up has been attached to the end of our responses.
* * *
**Comments by Referee #2:**

The authors present field measurements of isoprene, MACR and MVK at a forested mountain site in southern China. Relatively lower concentration levels of isoprene and higher ratios of (MVK+MACR)/isoprene than other forest sites were observed. The authors argued that the low isoprene levels were ascribed to the strong atmospheric oxidative capacity in the region, and a chemical box model constrained with observations further confirmed this argument. Overall, this paper presents new data of isoprene and its first-stage degradation intermediates in a unique environment with interactions of both biogenic and anthropogenic emissions. It is well organized and written. Therefore, this manuscript can be considered for publication after the following specific comments being properly addressed.

Response: We thank the referee for this overview. We have made changes to the manuscript based on the referee's helpful comments.
* * *
The authors used an MCM chemical box model to predict the concentrations of OH and $NO_3$ radicals. This model has been applied in several previous studies. However, there are still several issues that need to be clarified, and some additional modeling runs are needed to check the sensitivity of modelling results to these issues.

-**1.** Some previous studies have suggested that the MCM model may not work well for reproducing the $HO_x$ concentrations at high-BVOCs and low-$NO_x$ conditions, which seems to be the case of the study area in the present study. The authors may need to check the applicability of the MCM to the environmental condition at this forested mountain site.

Response: Many thanks and we agree with the referee that MCM needs to be improved and further optimized at pristine forest environments. And also as the OH mixing ratios modelled by PBM-MCM can only represent the levels at the site. Therefore, a parameterization method using measured aromatic hydrocarbons has been applied to estimate the regional mixing ratios of daytime OH radicals (Fig. 1 and Fig. 2) (Shiu et al., 2007). We think that this additional approach provides a good complement to the current model for evaluating the atmospheric oxidative capacity in the present forest regions. We have therefore added this point in the Section 2.4 and 3.3.1.

In the Section 2.4, Page 6:" Few previous studies have suggested that MCM may not work well for reproducing the OH concentrations in pristine forest environments (Kim et al., 2015). In addition, the OH mixing ratios modelled by the PBM-MCM can only represent the levels at the site. To provide a complement to the PBM-MCM, a widely-used parameterization method using measured aromatic hydrocarbons (i.e. BTEX, benzene, toluene, ethylbenzene, and m,p-xylene) was applied to estimate the regional mixing ratios of daytime OH (Shiu et al., 2007). The ratios of any two aromatics having the same emission sources but different photochemical reactivities can be used to measure photochemical oxidation (Parrish et al., 2007). This approach is based on three assumptions: (1) BTEX are removed from the atmosphere following pseudo-first-order kinetic; (2) no fresh BTEX are emitted to the air mass in the transport, and (3) the effects of horizontal and vertical mixing are similar for each compound. More details about the parameterization method are given in the Text S1."

In the Section 3.3.1, Page 9:"…To provide a complement to the PBM-MCM, regional mixing ratios of OH during 9:00-15:00 LST were calculated by a widely-used parameterization method using measured aromatic hydrocarbon ratios, i.e. toluene/benzene (T/B), ethylbenzene/benzene (E/B), and m,p-xylene/benzene (X/B) (Fig. S3). The average regional concentrations of OH during 9:00-15:00 LST

was 19.7 ± 2.3 × 10⁶ molecules cm⁻³, even higher than the modelled site-level OH of 11.7 ± 0.4 × 10⁶ molecules cm⁻³....".

[Figure]

Fig. 1: Scatterplots of the regional mixing ratios of OH during 09:00–15:00 LST derived from the toluene/benzene (OH$_{T/B}$), ethylbenzene/benzene (OH$_{E/B}$) and m,p-xylene/benzene (OH$_{X/B}$) ratios. The green line denotes a 1:1 relationship. Next to axes are the box and whisker plots of each result, and the pink dotted lines denote the mean values.

[Figure]

Fig. 2: Hourly variations of OH concentrations derived from the toluene/benzene ratio (OH$_{T/B}$), ethylbenzene/benzene ratio (OH$_{E/B}$), m,p-xylene/benzene ratio (OH$_{X/B}$) and PBM-MCM during 09:00 – 15:00 LST. Short-dashed lines denote the mean values of each method. Error bars indicate the 95% confidence interval.

*References:*

*Shiu, C. J., Liu, S. C., Chang, C. C., Chen, J. P., Chou, C. C. K., Lin, C. Y., and Young, C. Y.: Photochemical production of ozone and control strategy for Southern Taiwan, Atmos Environ, 41, 9324-9340, 10.1016/j.atmosenv.2007.09.014, 2007.*

*Kim, S., Kim, S. Y., Lee, M., Shim, H., Wolfe, G. M., Guenther, A. B., He, A., Hong, Y., and Han, J.: Impact of isoprene and HONO chemistry on ozone and OVOC formation in a semirural South Korean forest, Atmos Chem Phys, 15, 4357-4371, 10.5194/acp-15-4357-2015, 2015.*
* * *
-**2.** The NO$_2$ measurement analyzer used in this study may significantly overestimate for NO$_2$ at rural and remote sites. The authors are suggested to conduct more modeling analyses with artificially reduced NO$_2$ concentrations to examine the sensitivity of predicted OH and NO$_3$ to the input NO$_2$ data.

Response: Many thanks for spotting this, we have therefore conducted sensitivity analyses of modelled daytime OH and nighttime NO$_3$ with cutting NO$_2$ concentrations. The OH and NO$_3$ concentrations showed decreasing trends with the NO$_2$ reducing (Fig. 3). Assuming the average concentrations of daytime and nighttime NO$_2$ have been overestimated by 64.4 ± 2.9% and 62.4 ± 3.0% (Fig. 4), respectively, according to the results at a high-altitude mountain site in China (Xu et al., 2013), the mean daytime OH concentration at the present site would decrease from 7.3 ± 0.5 × 10$^6$ molecules cm$^{-3}$ to 5.5 ± 0.4 × 10$^6$ molecules cm$^{-3}$, with a reduction rate of 23.8 ± 6.2%. And for nighttime NO$_3$, the decreasing rate was 41.5 ± 5.2% (from 6.0 ± 0.5 × 10$^8$ molecules cm$^{-3}$ to 3.5 ± 0.3 × 10$^8$ molecules cm$^{-3}$). We have therefore added discussions about this point in the Section 2.3 and 3.3.3.

In the Section 2.3, Page 6: "…Sensitivity analyses were conducted for the model by varying the mixing ratios of NO$_2$.…"

In the Section 3.3.3, Page 8: "Three issues should be noted in applying PBM-MCM to evaluate the AOC in the present study. First, the NO$_2$ concentrations, an important input into PBM-MCM, may be significantly overestimated at this remote mountaintop site that receives a considerable amount of photochemically aged air (Xu et al., 2013). Thus we conducted sensitivity analyses of modelled OH and NO$_3$ with artificially reduced NO$_2$ concentrations for the period Aug.11−Aug.15 2016. The OH and NO$_3$ concentrations decrease with cutting NO$_2$ (Fig. S4). According to a recent study conducted at Mount Tai (Xu et al., 2013), we assumed the daytime and nighttime NO$_2$ measurements were overestimated by 64.4 ± 2.9% and 62.4 ± 3.0%, respectively. Thus the recalculated mean daytime OH concentration would decrease from 7.3 ± 0.5 × 10$^6$ molecules cm$^{-3}$ to 5.5 ± 0.4 × 10$^6$ molecules cm$^{-3}$, with a reduction rate of 23.8 ± 6.2%. And for nighttime NO$_3$, the reduction rate was 41.5 ± 5.2% (from 6.0 ± 0.5 × 10$^8$ molecules cm$^{-3}$ to 3.5 ± 0.3 × 10$^8$ molecules cm$^{-3}$).....".

[Figure]

Fig. 3: Sensitivity analysis of PBM-MCM modelled daytime OH and nighttime NO$_3$ with reduced NO$_2$ concentrations for the period Aug 11 – Aug 15 2016.

[Figure]

Fig. 4: Hourly modification factor of NO$_2$ during Jul 15–Aug 17 2016 at the Nanling site. The data in the figure are reproduced from the study conducted at a high-altitude mountain site (Mt. Tai) in central-eastern China by Xu et al. (2013).

*References:*

*Xu, Z., Wang, T., Xue, L. K., Louie, P. K. K., Luk, C. W. Y., Gao, J., Wang, S. L., Chai, F. H., and Wang, W. X.: Evaluating the uncertainties of thermal catalytic conversion in measuring atmospheric nitrogen dioxide at four differently polluted sites in China, Atmos Environ, 76, 221-226, 10.1016/j.atmosenv.2012.09.043, 2013.*
* * *
-**3.** The heterogeneous reactions of $N_2O_5$ play an important role in the nocturnal $NO_3$ chemistry. How does the PBM-MCM model present the $N_2O_5$ chemistry? What uptake coefficients of $N_2O_5$ onto particles were adopted in the model? Previous MCM modelling studies with addition of heterogeneous $N_2O_5$ chemistry have indicated the significant effect of this process on the simulation of photochemical processes (*Xue et al., 2014*). Additional modelling analyses are needed to examine the sensitivity of $NO_3$ to the $N_2O_5$ heterogeneous chemistry.

Response: We appreciate the referee's question and we agree that taking $N_2O_5$ heterogeneous chemistry into consideration would be very meaningfully. Unfortunately, we were not able to quantitatively take into account this important mechanism. We have admitted this limitation in the manuscript.

In the Section 2.3, Page 6: "It is noteworthy that … the heterogeneous process of $N_2O_5$ were not considered in the PBM-MCM.…..".

In the Section 3.3.3, Page10-11: "…Finally, the dinitrogen pentoxides ($N_2O_5$) that formed from the oxidation of $NO_2$ by $NO_3$ can be taken up onto aerosols via heterogeneous reactions, which is an important sink of $NO_2$ and $O_3$ at night and can compete with the reactions of $NO_3$ with isoprene (Xue et al., 2014b;Brown et al., 2016;Millet et al., 2016). Unfortunately, we were not able to quantitatively take into account this important mechanism in this study, and further studies are needed to make up this limitation."

*References:*
*Xue, L. K., Wang, T., Gao, J., Ding, A. J., Zhou, X. H., Blake, D. R., Wang, X. F., Saunders, S. M., Fan, S. J., Zuo, H. C., Zhang, Q. Z., and Wang, W. X.: Ground-level ozone in four Chinese cities: precursors, regional transport and heterogeneous processes, Atmos. Chem. Phys., 14, 13175-13188, 10.5194/acp-14-13175-2014, 2014b.*
*Millet, D. B., Baasandorj, M., Hu, L., Mitroo, D., Turner, J., and Williams, B. J.: Nighttime Chemistry and Morning Isoprene Can Drive Urban Ozone Downwind of a Major Deciduous Forest, Environ Sci Technol, 50, 4335-4342, 10.1021/acs.est.5b06367, 2016.*
*Brown, S. S., Dube, W. P., Tham, Y. J., Zha, Q. Z., Xue, L. K., Poon, S., Wang, Z., Blake, D. R., Tsui, W., Parrish, D. D., and Wang, T.: Nighttime chemistry at a high altitude site above Hong Kong, Journal of Geophysical Research-Atmospheres, 121, 2457-2475, 10.1002/2015jd024566, 2016.*
* * *
-**4.** Similarly, the HONO chemistry was also not well represented in the MCM model, but plays an important role in the OH simulation (*Xue et al., 2014*). Did the PBM-MCM model take this chemistry into account? If not, the authors may need to conduct some sensitivity analyses or at least mention the potential uncertainty of modeling results.

*Xue et al., Ground-level ozone in four Chinese cities: precursors, regional transport and heterogeneous processes, Atmos. Chem. Phys., 14, 13175- 13188, 2014.*

Response: Many thanks for spotting this. Since we did not measure the concentrations of HONO in the sampling periods, the average diurnal profiles of HONO observed at a background site (Hok Tsui) in Hong Kong in autumn 2012 (Zha, 2015) were applied to conduct sensitivity analyses. The results showed that daytime OH concentrations with HONO taken into account was 22 ± 19% higher than that without HONO (Fig. 5). Nonetheless, the results are not representative due to the lack of HONO measurements at the present site. Thus, certain uncertainties remain and we have therefore added discussions about this point in the Section 2.3 and Section 3.3.3.

In the Section 2.3, Page 6: "…Sensitivity analyses were conducted for the model by varying the mixing ratios of … and HONO….".

In the Section 3.3.3, Page 10: "…Second, a number of studies have shown that HONO plays an important role in daytime OH formation (Xue et al., 2014b). As the concentrations of HONO were not measured in the sampling periods, we therefore conducted sensitivity analyses by using a two-day (Aug 13 and Aug 15) dataset coupled with the average diurnal profiles of HONO observed at a background site (Hok Tsui) in Hong Kong in autumn 2012 (Zha, 2015). The results showed that daytime OH concentrations with HONO considered was 22 ± 19% higher than that without HONO (Fig. S5)….".

[Figure]

Fig. 5: Sensitivity analysis of PBM-MCM modelled daytime OH with and without HONO for the period Aug 13 and Aug 15 2016. The HONO data was obtained from the study conducted at a background site (Hok Tsui) in Hong Kong in autumn 2012 by Zha (2015).

*References:*

*Zha, Q.: Measurement of nitrous acid (HONO) and the implications to photochemical pollution, MPhil dissertation, Department of Civil and Environmental Engineering, The Hong Kong Polytechnic University, 2015.*

*Xue, L. K., Wang, T., Gao, J., Ding, A. J., Zhou, X. H., Blake, D. R., Wang, X. F., Saunders, S. M., Fan, S. J., Zuo, H. C., Zhang, Q. Z., and Wang, W. X.: Ground-level ozone in four Chinese cities: precursors, regional transport and heterogeneous processes, Atmos. Chem. Phys., 14, 13175-13188, 10.5194/acp-14-13175-2014, 2014b.*
* * *
-**5.** Sections 2.4 and 2.5: additional information about the calculation methods of the isoprene reaction time and air mass age, including principles and uncertainties, are required for readers to better understand and reproduce the results. The authors may also need to comment on the difference between reaction time and air mass age.

Response: Many thanks for spotting this. The description of the isoprene reaction time calculation have been modified and supplied in the Section 2.5 and Section 3.4. As we realized that the air mass age calculation was not closely related to the topic of the manuscript, this part has been removed out of the revised version.

In the Section 2.5, Page 7: "To check out the magnitude of isoprene oxidation, the initial isoprene was calculated using a "sequential reaction approach" based on the isoprene's oxidation mechanism and an empirical relationship between isoprene and its first-stage oxidation products (*i.e.* MVK and MACR) (Wolfe et al., 2016). This simplified parameterization method is based on four assumptions: (1) no fresh emissions of isoprene are introduced and isoprene emissions are constant during the process; (2) there were no additional sources of MVK and MACR apart from the oxidation of isoprene; (3) the processing time of the air mass are identical for all three compounds; and (4) only purely chemical reactions are included and the effects of turbulent mixing, horizontal convection and deposition are not accounted for. More description of the calculation is given in the Text S2."

In the Section 3.4, Page 11: "Fig. 6 shows the derived isoprene reaction times (IRT) from [MVK]/[isoprene] and [MACR]/[isoprene], respectively. IRT derived from the two ratios exhibit a significant linear correlation ($R^2$=0.91 and 0.90 for daytime and nighttime, respectively). The IRT derived from [MACR]/[isoprene] is 13% lower than that from [MVK]/[isoprene] on average, and we use the mean of these two values. The median and mean IRT during the day is 0.27 and 1.39 hr, respectively, with 4.10 and 4.49 hr at night. The median daytime reaction time (0.27 hr) of measured isoprene was slightly lower than the theoretical lifetime of isoprene (0.40 hr at 12-h daytime averaged [OH] = $8.0 \times 10^6$ molecules cm$^{-3}$). The short reaction time of isoprene indicates fast isoprene oxidation at this mountaintop site."

Response: Many thanks for spotting this, correction have been made to read "To investigate the atmospheric oxidative capacity (AOC) in forested high mountain areas adjacent to the photochemistry-active Pearl River Delta (PRD) region in southern China, one-month online observations of isoprene and its oxidation products methyl vinyl ketone (MVK) and methacrolein (MACR) were conducted at a national background station in Nanling Mountains in summer 2016.".
* * *
-**8.** Page 2, Lines 27-31: the oxidation of isoprene by OH radicals is very complex. MACR and MVK can be further oxidized to form MGLY and other secondary compounds. Here it would be helpful to add several sentences to briefly summarize the thorough oxidation chemistry of isoprene as well as the major knowledge gaps in understanding this chemistry.

Response: Many thanks, we have therefore added brief descriptions and major gaps in this paragraph: "…In the real ambient environment, the major competing reaction pathways include both NO- and HO$_2$-channels which dominate in polluted and pristine atmospheres, respectively (Paulot et al., 2009;Su et al., 2016). The relative importance of the two pathways varies with NO$_x$ (NO$_x$ = NO + NO$_2$) mixing ratios. Ambient measurements in pristine Amazon forests demonstrated that high OH concentrations often occur under high-isoprene and low-NO$_x$ (< 1 ppbv) conditions where OH regeneration contributes greatly to the oxidative capacity of the atmosphere (Lelieveld et al., 2008;Fuchs et al., 2013;Rohrer et al., 2014). Several recent studies have shown that small increases of NO$_x$ concentration above the background level can lead to a large change in the oxidative capacity and chemistry of the forest atmosphere (Liu et al., 2016;Su et al., 2016;Santos et al., 2018;Liu et al., 2018). In addition, the high OH-recycling efficiency is not unique to pristine forests, an important but different OH-recycling mechanism has been discovered in an isoprene-emitting forest suffering from heavy air pollution (Hofzumahaus et al., 2009). Thus, it is vital to understand the isoprene photochemistry under moderately polluted forest atmospheric conditions with high isoprene emissions and a broad range of NO$_x$ concentrations."

Response: Many thanks for spotting this, corrections have been made to read "This paper is structured as follows. Firstly, an overview of the meteorological and chemical conditions is given. Second, the measured concentrations and diurnal variations of isoprene and its oxidation products are presented. Then the estimated concentrations of daytime OH and nighttime $NO_3$ are presented and discussed in detail. And furthermore, the initial mixing ratios and atmospheric reaction time of isoprene were estimated. Finally, concluding remarks including a synthesis of current findings and some implications are presented. In this study, unexpected low isoprene levels and high (MVK+MACR)/isoprene ratios were observed. The subsequent theoretical calculations confirmed that the rapidly and highly isoprene oxidation might be attributable to a strong AOC in relation to the elevated regional complex air pollution in southern China."
* * *
-**11.** Page 4, Line 19: Southeast Asia

Response: Many thanks for spotting this – correction made.
* * *
-**12.** Change "atmospheric boundary layer (ABL)" to "planetary boundary layer (PBL)" throughout the manuscript.

Response: Many thanks, correction made.
* * *
-**13.** Page 5, Lines 18-19: it has been confirmed that this commercial $NO_x$ analyzer with a default molybdenum oxide converter can significantly overestimate for $NO_2$, especially in rural and remote areas such as the forested mountaintop in the present study. The authors may need clarify the detailed configuration of this $NO_x$ analyzer (e.g., the converters for converting $NO_2$ to NO) and state the uncertainty of $NO_2$ measurements if the MoO converter was used.

*Xu et al., Evaluating the uncertainties of thermal catalytic conversion in measuring atmospheric nitrogen dioxide at four differently polluted sites in China, Atmos. Environ., 76, 221-226, 2013.*

Response: We agree with the referee on the impact of molybdenum oxide converter may have on the measured $NO_2$ concentrations and we have added a discussion accordingly in the Section 2.2.2 and Section 3.1, respectively.

In the Section 2.2.2, Page 6: "$NO_2$ is converted to NO by a heated molybdenum converter before it can be measured by the chemiluminescence detection of NO. This method may cause an overestimation of $NO_2$ because the measured $NO_2$ probably includes some oxidized reactive nitrogen converted by the heated molybdenum (Xu et al., 2013). Thus, the $NO_2$ concentrations given below are considered as the upper limits of their actual values."

In the Section 3.1, Page 8: "Since the $NO_2$ concentrations measured by the molybdenum oxide converter technique can be significantly overestimated in areas far away from fresh $NO_x$ emission sources (Xu et al., 2013), we corrected the observed $NO_2$ by adopting the hourly modification factors (Fig. S1) obtained at Mount Tai (1,533 m a.s.l.) in central-eastern China (Xu et al., 2013). The modified $NO_2$ (889 ± 27 pptv) was 1.1−2.5 times (1.8 ± 0.3) lower than that observed."

The "polarPlot" technique was therefore used for source identification (Fig. 7). The red dotted sectorial domains in the figure are interpreted as the regional transport interference as the concentrations of species increase with increasing wind speed. The high levels of species at high wind speeds most likely came from the nearby urban centers. Therefore, measurements that are deemed to be affected by regional transport are all excluded from the dataset in the analysis.

We have therefore added a discussion accordingly in Section 3.2, Page 9: "…with little transported from anthropogenic sources in neighbor cities at night (see discussion in Text S3)."

[Figure]

Fig. 6: (a) Location of the Nanling site, Dinghu Mountain site, Hok Tsui site, Guangzhou and Hong Kong. The Nanling site is 174 km northeast to the Dinghu Mountain site 178 km northwest to Guangzhou. Red outlined domain represent the Pearl River Delta region. (b) Map showing the nearest urban centers (Yangshan County, Ruyuan County, Lechang City, Lianzhou City, Shaoguan City and Yingde City) around the site.

[Figure]

[Figure]

Fig. 7: Daytime and nighttime polarplots of MVK and MACR during the sampling period (July 15–August 17 2016). Concentrations varied by wind speed (ws, unit in m/s) and wind direction. Red dotted sectorial domains represent the interferences of regional transport from nearby urban centers.

* * *
-**18.** Page 10, Line 17, "the observed relationship of observed MVK/isoprene": delete one "observed"

Response: Many thanks, correction made.
* * *
-**19.** Table 1: provide the standard deviation of the average concentrations if available.

Response: Many thanks, done.

**Comments by Referee #3:**

Gong et al. presents their results of online observations of isoprene and its first-stage oxidation products MVK and MACR in summer 2016 at a remote, high-altitude mountain forest site to the north of the air-polluted PRD region in southern China. They found that the isoprene level was significantly lower and attributed it to the strong regional atmospheric oxidative capacity. The PBM-MCM model was used to estimate the OH and $NO_3$ concentrations to support their assumptions. The paper is well written and organized. The reviewer would recommend the manuscript for publication after some specific comments.

Response: We thank the referee for this overview. We have made changes to the manuscript based on the referee's helpful comments.
* * *
-**1.** $O_3$, $OH_x$, PAN, and $NO_3$ are indicators of atmospheric oxidative capacity. Since OH and $NO_3$ were not determined in the observation, the observed $O_3$ concentration is a more powerful tool to express the atmospheric oxidative capacity. The diurnal variations of $O_3$ peaked at 20:00 is very interesting, because the changing trends of $O_3$ and sun radiation were not accordant. The temporal variations of $O_3$ also show different trends during the observation. Could regional transport contribute $O_3$ to the measurement site? The authors had better add more discussion on the variations of $O_3$ concentration.

Response: Many thanks for spotting this. We noticed that the high levels of $O_3$, as well as its distinct diurnal variations, were observed at the site, and we are preparing another paper to discuss these points in details. In this revised manuscript, discussions about the high abundances of $O_3$ at the site has been added in the Section 3.1. As for the interesting $O_3$ diurnal variations, a number of studies have observed a similar pattern at mountaintops, that is, high concentrations at night and low concentrations during the daytime (Gallardo et al., 2000;Gao et al., 2017). Some previous studies have concluded that the factors closely related to the distinct $O_3$ diurnal patterns at mountaintops were the boundary layer diurnal cycles (Gao et al., 2017), the mountain-valley breezes (Zaveri et al., 1995;Yang et al., 2012;Cristofanelli et al., 2013), the regional transport effects (Naja et al., 2003;Li et al., 2008;Zhang et al., 2015;Gao et al., 2017), the location and altitude of a mountain (Chevalier et al.,

2007;Monteiro et al., 2012), and ozone vertical distributions (Forrer et al., 2000;Zellweger et al., 2003;Gheusi et al., 2011). Overall, the distinct diurnal variations in $O_3$ concentrations are the result of a combination of various physical and chemical processes. This point has been added to the Section 3.2.

In the Section 3.1, Page 7-8:" Similar as previous mountaintop studies, a distinct $O_3$ diurnal variation featured with high levels at night and low levels during the daytime (Zaveri et al., 1995;Gao et al., 2017) was observed in this study, most likely due to a combination of various physical and chemical processes (*e.g.* boundary layer diurnal cycles, mountain-valley breezes, regional transport, photochemical reactions)."

In the Section 3.2, Page 8:" High abundances of $O_3$ at this site likely indicate strong oxidizing power of the present forest atmospheres."

Response: Many thanks for spotting this, the correction has been made to read "To investigate the atmospheric oxidative capacity (AOC) in forested high mountain areas adjacent to the photochemistry-active Pearl River Delta (PRD) region in southern China, one-month online observations of isoprene and its oxidation products methyl vinyl ketone (MVK) and methacrolein (MACR) were conducted at a national background station in Nanling Mountains in summer 2016.".
* * *
-**4.** Page 4, Line 32, the specifications of the Teflon filter should be clarified.

Response: Many thanks for spotting this, corrections have been made to read "…a Teflon filter (0.25 µm pore size, 47 mm OD, Millipore, USA) was placed in front of the sample inlet."
* * *
-**5.** Page 5, Line 21-23, it is confused that "daily" and "every two days". Secondly, it seems that $SO_2$, $NO_x$, and CO analyzers are usually calibrated with domestic standard gases which are not NIST-traceable. The NIST-traceable standard was only applied to calibrate $O_3$ analyzer.

Response: Many thanks for spotting this mistake, corrections have been made to read "
[revised manuscript text omitted]

aromatic hydrocarbons regarding their universal anthropogenic sources. Because of their atmospheric
lifetimes of up to several days (Table S1), BTEX can undergo long-range transport. The ratio of two
aromatics that share common emission sources but with different reactivities with hydroxyl radical (OH)
can be used as a measure of photochemical oxidation by OH (Parrish et al., 2007;Shiu et al., 2007). Thus,
BTEX provide particularly promising capabilities for following photochemical processing on timescales of
hours to days. In this study, we chose three pairs of aromatic species: toluene/benzene,
ethylbenzene/benzene, and m,p-xylene/benzene.

During daytime, decreasing of BTEX levels along transport path owes mainly to atmospheric dilution and
reaction with OH. The chemical removal of BTEX by OH can be expressed as followings:

$Benzene + OH \rightarrow$ products $\qquad k_{benzene,OH} = 2.3 \times 10^{-12} \, e^{-190/T}$ $\qquad$ (R1)

$Toluene + OH \rightarrow$ products $\qquad k_{Toluene,OH} = 1.8 \times 10^{-12} \, e^{340/T}$ $\qquad$ (R2)

$Ethylbenzene + OH \rightarrow$ products $\quad k_{Ethylbenzene,OH} = 7.0 \times 10^{-12}$ $\qquad$ (R3)

$m,p - Xylene + OH \rightarrow$ products $\quad k_{m,p-Xylene,OH} = 1.89 \times 10^{-11}$ $\qquad$ (R4)

Thus, mixing ratios of BTEX at the sampling time can be expressed as follows, for example, in the case of
benzene,

$$[B]_t = [B]_0 \times e^{-[OH] \times k_{B,OH} \times t} \times f_{d,B} \hspace{4cm} \text{(Eq.1)}$$

where $[B]_0$ and $[B]_t$ represents the mixing ratios of measured benzene at the start and after transport time $t$

that air mass spent in the atmosphere, respectively. $k_{B,OH}$ represents the temperature dependent reaction rate coefficient of benzene with OH, which was taken from the IUPAC database (http://iupac.pole-ether.fr/)

(Atkinson et al., 2006). $[OH]$ represents the regional concentrations of OH. $f_{d,B}$ represents the dilution factor of benzene in the atmosphere.

Toluene, ethylbenzene and m,p-xylene react faster with OH than benzene. In this study, we assuming the rates of turbulent mixing and horizontal convection are similar for BTEX. Therefore, during the transport time $\Delta t$, the dilution factor of BTEX are the same. Then rearranging Eq.1 and extending this analysis to

BTEX will yield the following equations.

$$[OH]_{T/B} = \frac{1}{t \times (k_{T,OH} - k_{B,OH})} \times \left[ \ln\left(\frac{[T]}{[B]}\right)_0 - \ln\left(\frac{[T]}{[B]}\right)_t \right] \hspace{2cm} \text{(Eq.2)}$$

$$[OH]_{E/B} = \frac{1}{t \times (k_{E,OH} - k_{B,OH})} \times \left[ \ln\left(\frac{[E]}{[B]}\right)_0 - \ln\left(\frac{[E]}{[B]}\right)_t \right] \hspace{2cm} \text{(Eq.3)}$$

$$[OH]_{X/B} = \frac{1}{t \times (k_{X,OH} - k_{B,OH})} \times \left[ \ln\left(\frac{[X]}{[B]}\right)_0 - \ln\left(\frac{[X]}{[B]}\right)_t \right] \hspace{2cm} \text{(Eq.4)}$$

Where $[OH]_{T/B}$, $[OH]_{E/B}$, and $[OH]_{X/B}$ represent the estimated regional mixing ratios of OH by three aromatic ratios. It should be noted that the data included only daytime observations between 09:00–15:00

LST when the aromatic ratios presented a smoothly decreasing trend (Fig. S3). In addition, the X/B ratio may not work well in comparison with T/B and E/B owing to the relatively short lifetime of m,p-xylene (Table S1). Indeed, results derived from X/B are significantly lower than those from T/B and E/B, and we use the mean values derived by T/B and E/B in this study. The scatterplots of the regional mixing ratios of

OH derived from the three ratios were presented in the Fig. S8, and the diurnal patterns were presented in the Fig. S9.

**Text S2.** Calculation of initial isoprene

The first-stage oxidation of isoprene (ISO), MVK and MACR by reaction with OH during the day, and with

$NO_3$ at night, can be described by the following reactions:

$ISO + OH \rightarrow y_{MACR,OH} \, MACR + y_{MVK,OH} \, MVK$ $\hspace{1.5cm} k_{ISO,OH} = 2.7 \times 10^{-11} \, e^{390/T} \hspace{2cm} \text{(R5)}$

$MVK + OH \rightarrow products$ $\hspace{4cm} k_{MVK,OH} = 2.6 \times 10^{-12} \, e^{610/T} \hspace{2cm} \text{(R6)}$

$MACR + OH \rightarrow products$ $\hspace{3.5cm} k_{MACR,OH} = 8.0 \times 10^{-12} \, e^{380/T} \hspace{2cm} \text{(R7)}$

$ISO + NO_3 \rightarrow y_{MACR,NO_3} \, MACR + y_{MVK,NO_3} \, MVK$ $\hspace{1cm} k_{ISO,NO_3} = 2.95 \times 10^{-12} \, e^{-450/T} \hspace{1.5cm} \text{(R8)}$

$MVK + NO_3 \rightarrow$ products                               $k_{MVK,NO_3} < 6.0 \times 10^{-16}$                     (R9)

$MACR + NO_3 \rightarrow$ products                         $k_{MACR,NO_3} = 3.4 \times 10^{-15}$                (R10)

Where $k$ is the temperature dependent reaction rate coefficients taken from the IUPAC database (http://iupac.pole-ether.fr/) (Atkinson et al., 2006). $y$ are yields of MVK and MACR from isoprene reaction with OH and NO$_3$. $T$ is the temperature (unit in K).

Then, the product/parent ratios, $i.e.$ [MVK]/[ISO] and [MVK]/[ISO], can be calculated as a function of the rate constants ($k$), yield ($y$), reaction time ($\Delta t$) and radical concentration ([$OH$] and [$NO_3$]):

$$\left(\frac{[MVK]}{[ISO]}\right)_D = \frac{y_{MVK,OH}\, k_{ISO,OH}}{(k_{MVK,OH} - k_{ISO,OH})} \times \left[1 - e^{(k_{ISO,OH} - k_{MVK,OH})\,[OH]\,\Delta t}\right]$$     (Eq.5)

$$\left(\frac{[MACR]}{[ISO]}\right)_D = \frac{y_{MACR,OH}\, k_{ISO,OH}}{(k_{MACR,OH} - k_{ISO,OH})} \times \left[1 - e^{(k_{ISO,OH} - k_{MACR,OH})\,[OH]\,\Delta t}\right]$$     (Eq.6)

$$\left(\frac{[MVK]}{[ISO]}\right)_N = \frac{y_{MVK,NO_3}\, k_{ISO,NO_3}}{(k_{MVK,NO_3} - k_{ISO,NO_3})} \times \left[1 - e^{(k_{ISO,NO_3} - k_{MVK,NO_3})\,[NO_3]\,\Delta t}\right]$$     (Eq.7)

$$\left(\frac{[MACR]}{[ISO]}\right)_N = \frac{y_{MACR,NO_3}\, k_{ISO,NO_3}}{(k_{MACR,NO_3} - k_{ISO,NO_3})} \times \left[1 - e^{(k_{ISO,NO_3} - k_{MACR,NO_3})\,[NO_3]\,\Delta t}\right]$$     (Eq.8)

Where [$ISO$], [$MVK$] and [$MACR$] represent the observed isoprene, MVK and MACR. [$OH$] and [$NO_3$]

represent daytime OH and nighttime NO$_3$, respectively. $D$ and $N$ represent daytime and nighttime periods, respectively. $\Delta t$ is the atmospheric reaction time of isoprene, representing the time of isoprene in the atmosphere between emission and detection.

Initial isoprene [$ISO$]$_i$, the total isoprene emissions that have been released into the sample air masses, can be effectively calculated via reverse integration of isoprene's first-stage oxidation:

$$[ISO]_{i,D} = [ISO] \times e^{(k \times [OH] \times \Delta t)}$$     (Eq.9)

$$[ISO]_{i,N} = [ISO] \times e^{(k \times [NO_3] \times \Delta t)}$$     (Eq.10)

By determining the concentration of radical ($i.e.$ OH and NO$_3$) and atmospheric reaction time of isoprene, the initial isoprene can be calculated.

Two issues arise when applying this "back-of-the-envelope" method to the present study (Wolfe et al.,

2016). First, the yields from OH-initiated isoprene oxidation are a nonlinear function of nitrogen oxide (NO). Previous applications of this method (de Gouw, 2005;Roberts et al., 2006;Stroud et al., 2001;Karl et al., 2007;Kuhn et al., 2007) have assumed lab-derived high-NO yields of 0.33 and 0.23 for MVK and

MACR, respectively (Atkinson and Arey, 2003a), but this may not be appropriate in the present study. Thus we chose the yields derived from the latest Master Chemical Mechanism (MCM) v3.3.1 (Jenkin et al.,

2015). The resulting yield curves are interpolated to observed NO mixing ratios (Fig. S7). The yields from

NO$_3$-initiated isoprene oxidation are constants (Table S1). Second, the implied initial isoprene is depended on the mixing ratios of radical and the atmospheric reaction time of isoprene. Thus in this study we introduce the term "exposure" (de Gouw, 2005;Jimenez et al., 2009;Wolfe et al., 2016) defined here as the product of radical concentration and reaction time ($Exposure = [Radical] \times \Delta t$). Exposures can be obtained by the following two-step process.

First, for any given exposure, a daughter/parent ratio can be expected based on Eq. (5−8), and the theoretical line of [MVK]/[isoprene] versus [MACR]/[isoprene] can be depicted (Stroud et al., 2001;Apel et al., 2002;Roberts et al., 2006;Guo et al., 2012;Wolfe et al., 2016). The range of exposure thus can be derived by comparing the expected daughter/parent ratio with observed data. Then the range of the ratio of initial over observed isoprene ($ISO_i/ISO_o$) can be calculated based on Eq. (9−10). Fig. S10 compares the relationship of measured data against theoretical trends predicted by the sequential reaction calculation for the daytime and nighttime hours. It can be seen that the observed [MVK]/[isoprene] versus [MACR]/[isoprene] exhibit a tight linear correlation ($R^2$=0.68 and 0.72 for daytime and nighttime periods, respectively). The measured data fit the predicted line well, although most of the measured data are above the predicted line, which is consistent with some previous studies (Stroud et al., 2001;Apel et al., 2002;Guo et al., 2012). Apart from the uncertainties mentioned in the Section 2.5 of the manuscript, the additional sources of MVK and MACR from isoprene oxidation by daytime $NO_3$ (Xue et al., 2016) and nighttime OH (Lu et al., 2014) which both were not taken into account in the present study might also be the causes. The theoretical slope agrees well with observations, indicating OH exposures of $0.1-12 \times 10^6$ molecules $cm^{-3}$ h and $NO_3$ exposures of $4-28 \times 10^8$ molecules $cm^{-3}$ h for daytime and nighttime periods, respectively.

Second, detailed profiles of exposure can be directly calculated from the observed daughter/parent ratios by inverting Eq. (5−8):

$$EXPO_{OH,MVK} = \frac{\ln\left(1-\frac{[MVK]}{[ISO]} \times \frac{k_{MVK,OH}-k_{ISO,OH}}{y_{MVK,OH} \, k_{ISO,OH}}\right)}{(k_{ISO,OH}-k_{MVK,OH})} \tag{Eq.11}$$

$$EXPO_{OH,MACR} = \frac{\ln\left(1-\frac{[MACR]}{[ISO]} \times \frac{k_{MACR,OH}-k_{ISO,OH}}{y_{MACR,OH} \, k_{ISO,OH}}\right)}{(k_{ISO,OH}-k_{MACR,OH})} \tag{Eq.12}$$

$$EXPO_{NO_3,MVK} = \frac{\ln\left(1-\frac{[MVK]}{[ISO]} \times \frac{k_{MVK,NO_3}-k_{ISO,NO_3}}{y_{MVK,NO_3} \, k_{ISO,NO_3}}\right)}{(k_{ISO,NO_3}-k_{MVK,NO_3})} \tag{Eq.13}$$

$$EXPO_{NO_3,MACR} = \frac{\ln\left(1-\frac{[MACR]}{[ISO]} \times \frac{k_{MACR,NO_3}-k_{ISO,NO_3}}{y_{MACR,NO_3} \, k_{ISO,NO_3}}\right)}{(k_{ISO,NO_3}-k_{MACR,NO_3})} \tag{Eq.14}$$

Where $EXPO_{OH, MVK}$, $EXPO_{OH, MACR}$, $EXPO_{NO_3, MVK}$, and $EXPO_{NO_3, MACR}$ represent the derived exposures from [MVK]/[isoprene] and [MACR]/[isoprene] for daytime and nighttime periods.

Calculated daytime OH exposures and nighttime $NO_3$ exposures range from $1.0 \times 10^5$ to $1.3 \times 10^7$ molecules $cm^{-3}$ h and $3.5 \times 10^8$ to $3.2 \times 10^9$ molecules $cm^{-3}$ h, respectively (Fig. S11). The OH and $NO_3$ exposures derived from two ratios exhibit a good linear correlation ($R^2$=0.63 and 0.70 for OH and $NO_3$, respectively), and results derived from [MACR]/[isoprene] are 4% and 18% lower than those from

[MVK]/[isoprene] on average, respectively, and we use the mean of these two values. The median and mean OH exposure is 1.9 and $2.5 \times 10^6$ molecules cm$^{-3}$ h, respectively. For NO$_3$ exposure, the median and mean value is close (15.8 and $16.2 \times 10^8$ molecules cm$^{-3}$ h, respectively).

**Text S3.** **Data processing, graph plotting and graphical source identification by R**

In this study, the open source R package "openair" was utilized for data processing and graph plotting (Carslaw and Ropkins, 2012;Carslaw, 2015). Specifically, for those dedicated functions used, the

"transform", "selectByDate", "merge" and "subset" functions were used to calculate and filter the data. The

"quantile", "summary" and "t.test" functions were used to do statistical analysis. The "timePlot" function was used to plot time series of measured species. The "plot" function was used to plot scatter and diagram diurnal variations. In particular, the "polarPlot" function was used for source identification.

The lifetimes of MVK and MACR by OH loss is 1.9 and 1.0 hours, respectively, assuming 12-h daytime

OH = $8.0 \times 10^6$ molecules cm$^{-3}$ (Table S1). The average daytime wind speed was $3.9 \pm 0.2$ m s$^{-1}$ at the site, and the distance between the sampling sit and the nearest urban center is 38 km (Fig. S12), the air parcel from upwind locations would spend about 2.7 hours to arrive at the sampling site. This is enough time for the depletion of MVK and MACR along the traveling path during the daytime. However, the nighttime chemical oxidation of MVK and MACR was slow, with lifetimes of MVK and MACR by NO$_3$ oxidation of

0.5 years and 72 hours, respectively, assuming 12-h nighttime NO$_3$ = $5.0 \times 10^8$ molecules cm$^{-3}$ (Table S1).

Apart from biogenic sources, the anthropogenic sources of MVK and MACR, *e.g.* motor vehicles, biomass burning and industrial sources, have been reported by some previous studies (Borbon et al., 2001;Wagner and Kuttler, 2014;Hsieh et al., 2016;Diao et al., 2016). Therefore, the regional transport of MVK- or

MACR- laden air could affect the observed nighttime (MVK+MACR)/isoprene ratios at the site.

Hence, the "polarPlot" technique was used for source identification in this study. In "polarPlot" drawing, the species concentrations are shown to vary by wind speed and wind direction, and plots are shown as a continuous surface, and the surfaces are calculated through modeling using smoothing techniques. These plots have been proved to be useful for the quick gaining of a graphical impression of potential sources'

influences at a location in recent publications that describe or use the technique (Valach et al., 2014;Chang et al., 2017). Fig. S13 shows the polarplots of isoprene, MVK and MACR during the study. During the sampling periods, the air masses reaching the site were mainly from the southwest and northeast directions.

The red dotted sectorial domains are interpreted as the regional transport interference as the concentrations of species increase with increasing wind speed. The high levels of species at high wind speeds most likely came from the nearby urban centers. Therefore, measurements that are deemed to be affected by regional transport are all excluded from the dataset in the analysis.

**Tables**

**Table S1.** Rate constants and lifetime for isoprene, MVK and MACR, and yields of MVK and MACR from the isoprene reactions.

| Compound | Rate constants [a] with | | | Yield [b] from isoprene reaction with / (lifetime [b] due to) | | |
|---|---|---|---|---|---|---|
| | OH | $NO_3$ | $O_3$ | OH | $NO_3$ | $O_3$ |
| Isoprene | $1.0\times10^{-10}$ | $7.0\times10^{-13}$ | $1.3\times10^{-17}$ | - / 0.4 h | - / 0.8 h | - / 24 h |
| MVK | $2.0\times10^{-11}$ | $3.2\times10^{-16}$ | $5.2\times10^{-18}$ | 0.33 / 1.9 h | 0.035 / 0.5 yr | 0.16 / 61 h |
| MACR | $2.9\times10^{-11}$ | $3.7\times10^{-15}$ | $1.2\times10^{-18}$ | 0.23 / 1.0 h | 0.035 / 72 h | 0.41 / 10 d |
| Benzene | $1.2\times10^{-12}$ | $<3.0\times10^{-17}$ | $<1.0\times10^{-21}$ | - / 9.5 d [c] | - / >4 yr | - / >4.5 yr |
| Toluene | $5.6\times10^{-12}$ | $7.8\times10^{-17}$ | $<1.0\times10^{-21}$ | - / 2.1 d [c] | - / 1.8 yr | - / >4.5 yr |
| Ethylbenzene | $7.0\times10^{-12}$ | $<6.0\times10^{-16}$ | $<1.0\times10^{-21}$ | - / 1.7 d [c] | - | - |
| m,p-Xylene | $1.9\times10^{-11}$ | $3.8\times10^{-16}$ | $<1.0\times10^{-21}$ | - / 0.6 d [c] | - / 0.5 yr | - / >5 yr |

[a] The 298 K rate constants (unit in $cm^3\,molecule^{-1}\,s^{-1}$) are taken from Atkinson and Arey (2003b), Atkinson et al. (2006) and
IUPAC database (http://iupac.pole-ether.fr/).
[b] (Atkinson and Arey, 2003a, b) and references therein. Lifetime calculated at 298 K using the following: for OH radical
reactions, a 12-h daytime average of $8.0 \times 10^6$ molecules $cm^{-3}$; for $NO_3$ radical reactions, a 12-h nighttime average of $5.0 \times 10^8$
molecules $cm^{-3}$; and for $O_3$, a 24-h average of $1.0 \times 10^{12}$ molecules $cm^{-3}$.
[c] For a 12-h daytime average OH radical concentration of $2.0 \times 10^6$ molecules $cm^{-3}$.

**Figures**

[Figure]

**Fig. S1:** Hourly modification factor of $NO_2$ during Jul 15 – Aug 17 2016 at the Nanling site. The data in the figure are reproduced from the study conducted at a high-altitude mountain site (Mt. Tai) in central-eastern China by Xu et al. (2013).

[Figure]

**Fig. S2:** Diurnal variation of the Planetary Boundary Layer (PBL) height (data provided by Real-time
Environmental Applications and Display System, https://ready.arl.noaa.gov/READYamet.php, unit in m) at Nanling
site (1,690 m a.s.l.) during Jul 15–Aug 17 2016. Error bars indicate the 95% confidence interval.

[Figure]

**Fig. S3:** Diurnal variations of toluene/benzene, ethylbenzene/benzene and m,p-xylene/benzene ratios at Nanling site
during Jul 15–Aug 17 2016. Shaded regions denote the nighttime periods. Error bars indicate the 95% confidence
interval. Blue boxed area denotes periods for the calculating of regional mixing ratios of daytime OH.

[Figure]

**Fig. S4:** Sensitivity analysis of PBM-MCM modelled daytime OH and nighttime NO₃ with reduced NO₂

concentrations for the period Aug 11 − Aug 15 2016.

[Figure]

**Fig. S5:** Sensitivity analysis of PBM-MCM modelled daytime OH with and without HONO for the period Aug 13

and Aug 15 2016. The HONO data was obtained from the study conducted at a background site (Hok Tsui) in Hong

Kong in autumn 2012 by Zha (2015).

[Figure]

**Fig. S6:** Comparison of observed and initial isoprene mixing ratios at night. Green dashed lines denote slopes for different ratios of initial to observed isoprene.

[Figure]

**Fig. S7:** Molar yields of the main first-generation products (MVK and MACR) of the OH-initiated oxidation of isoprene as a function of NO mixing ratio, at 298 K, as represented in MCM v3.3.1 (Jenkin et al., 2015). Thin green vertical lines denote, from left to right, the 1st, 5th, 50th, 95th and 99th percentiles of hourly NO observed during the present study.

[Figure]

**Fig. S8:** Scatterplots of the regional mixing ratios of OH during 09:00 – 15:00 LST derived from the toluene/benzene ($OH_{T/B}$), ethylbenzene/benzene ($OH_{E/B}$) and m,p-xylene/benzene ($OH_{X/B}$) ratios. The green line denotes a 1:1 relationship. Next to axes are the box and whisker plots of each result, and the pink dotted lines denote the mean values.

[Figure]

**Fig. S9:** Average hourly variations of the regional concentrations of OH derived from the toluene/benzene ratio ($OH_{T/B}$), ethylbenzene/benzene ratio ($OH_{E/B}$) and m,p-xylene/benzene ratio ($OH_{X/B}$) and the site-level OH modelled by PBM-MCM between 09:00 and 15:00 LST.

[Figure]

**Fig. S10:** Ranges of exposure derived by the progression of product/parent ratios (*i.e.* [MVK]/[isoprene] and

[MACR]/[isoprene], unit: molecules cm$^{-3}$ / molecules cm$^{-3}$). Red circles and blue crosses show the observed ratios for the daytime and nighttime measurements, respectively. The red solid and blue dashed lines are the theoretical product/parent ratios of isoprene sequential reaction scheme calculation. The numbers next to the line indicate the theoretical exposures (the product of radical concentration and reaction time) corresponding to any given product

−parent relationship.

[Figure]

**Fig. S11:** Scatter plots of exposures derived from observed [MVK]/[isoprene] versus that from [MACR]/[isoprene].

The unit of OH exposure and NO$_3$ exposure is $10^6$ molecules cm$^{-3}$ h and $10^8$ molecules cm$^{-3}$ h, respectively.

[Figure]

**Fig. S12:** **(a) Location of the Nanling site, Dinghu Mountain site, Hok Tsui site, Guangzhou and Hong Kong. The**

**Nanling site is 174 km northeast to the Dinghu Mountain site and 178 km northwest to Guangzhou. Red outlined**

**domain represent the Pearl River Delta region. (b) Map showing the nearest urban centers (Yangshan County,**

**Ruyuan County, Lechang City, Lianzhou City, Shaoguan City and Yingde City) around the site.**

[Figure]

**Fig. S13:** Daytime and nighttime polarplots of isoprene, MVK and MACR during the sampling period (Jul 15–Aug

17 2016). Concentrations varied by wind speed (ws, unit in m/s) and wind direction. Red dotted sectorial domains represent the interferences of regional transport from nearby urban centres.